# Vacuolar sterol β-glucosidase EGCrP2/Sgl1 deficiency in *Cryptococcus neoformans*: Dysfunctional autophagy and Mincle-dependent immune activation as targets of novel antifungal strategies

Takashi Watanabe[1,2☯], Masayoshi Nagai[1,3☯], Yohei Ishibashi[1☯*], Mio Iwasaki[1], Masaki Mizoguchi[1], Masahiro Nagata[4], Takashi Imai[4], Koichi Takato[5], Akihiro Imamura[5,6], Yoshimitsu Kakuta[1], Takamasa Teramoto[1], Motohiro Tani[7,8], Junko Matsuda[2], Hideharu Ishida[5,6], Sho Yamasaki[4,9,10], Nozomu Okino[1], Makoto Ito[1]*

1 Department of Bioscience and Biotechnology, Faculty of Agriculture, Kyushu University, Fukuoka, Japan, 2 Department of Pathophysiology and Metabolism, Kawasaki Medical School, Kurashiki, Japan, 3 Department of Medical Biochemistry, Graduate School of Medicine, Osaka Metropolitan University, Osaka, Japan, 4 Department of Molecular Immunology, Research Institute for Microbial Diseases, Osaka University, Osaka, Japan, 5 Department of Applied Bioorganic Chemistry, Gifu University, Gifu, Japan, 6 Institute for Glyco-core Research (iGCORE), Gifu University, Gifu, Japan, 7 Department of Chemistry, Faculty of Science, Kyushu University, Fukuoka, Japan, 8 Faculty of Applied Biological Sciences, Gifu University, Gifu, Japan, 9 Laboratory of Molecular Immunology, Immunology Frontier Research Center, Osaka University, Osaka, Japan, 10 Center for Infectious Disease Education and Research, Osaka University, Osaka, Japan

☯ These authors contributed equally to this work.
* makotoi@agr.kyushu-u.ac.jp (MI); ishibashiyo@agr.kyushu-u.ac.jp (YI)

## Abstract

*Cryptococcus neoformans* (*Cn*) is a fungal pathogen responsible for cryptococcal meningitis, which accounts for 15% of AIDS-related deaths. Recent studies have shown that the absence of sterol β-glucosidase (EGCrP2, also known as Sgl1) in *Cn* significantly attenuates its virulence in a mouse infection model. However, the mechanisms underlying this virulence attenuation remain unclear. In this study, we observed a significant increase in dead cells after 3 days of culture of *SGL1*-deficient *Cn* (*sgl1Δ*, KO) at 37°C, compared with wild-type (WT) and *SGL1*-reconstituted *Cn* (*sgl1Δ::SGL1*, RE). qPCR analysis of WT, KO, and RE strains indicated that autophagy-related genes (*ATG*s) were significantly downregulated in KO strain. Atg8-dependent GFP translocation to the vacuole was significantly delayed in KO strain under starvation conditions. This autophagy dysfunction was identified as the primary cause of the increased cell death observed in KO strain under nitrogen starvation conditions at 37°C. EGCrP2/Sgl1 is predominantly localized in the vacuoles of *Cn*, and its deletion results in the accumulation of not only ergosterol β-glucoside (EG), as previously reported, but also acylated EGs (AEGs). AEGs were much more potent than EG in activating the C-type lectin receptor Mincle in mice, rats, and humans. AEGs were released from KO strain via extracellular vesicles

**Data availability statement:** The authors declare that all data supporting this study's results are available in the article and its Supplementary Appendix.

**Funding:** This research was supported by AMED under grant number JP21gm0910010 (MI, NO, SY). The funders had no role in study design, data collection and analysis, the decision to publish, or the preparation of the manuscript.

**Competing interests:** The authors have declared that no competing interests exist.

(EVs). Chemically synthesized 18:1-EG and EVs derived from KO strain, but not WT or RE strains, enhanced cytokine production in murine and human dendritic cells. AEG-dependent cytokine production was markedly reduced in dendritic cells from Mincle-deficient mice, and the number of KO strain in lung tissue from Mincle-deficient mice was substantially higher than wild-type mice on day 3 after infection. Intranasal administration of acylated sitosterol β-glucoside increased Mincle expression and cytokine production and reduced the *Cn* burden in lung tissue of *Cn*-infected mice. These findings suggest that autophagy dysfunction in KO strain and the host innate immune response via the AEG-dependent Mincle activation are critical in reducing *Cn* virulence in mice.

## Author summary

The WHO recently flagged *Cn* as one of the most dangerous pathogenic fungi, highlighting the urgent need for new therapeutic targets. EG, the major glycolipid of *Cn,* is catabolized by EGCrP2/Sgl1, widely distributed in the fungal kingdom, including all critical pathogenic fungi defined by WHO. This study elucidates how genetic disruption of EGCrP2/Sgl1 significantly reduces the virulence of *Cn* in mice from both the pathogen and host perspectives. In KO strain, EG accumulates in the vacuole and is converted to its acylated derivatives, AEGs. We found that accumulation of EG and AEGs leads to increased lethality of KO strain under nutrient starvation at 37°C due to defective autophagy. AEGs are transported out of fungal cells via extracellular vesicles and act as a ligand for the C-type lectin Mincle, expressed on host dendritic cells and macrophages. AEG-dependent Mincle activation contributes to the clearance of KO strain from mice at an early stage of infection. Our research indicates that EGCrP2/Sgl1 is a promising target for antifungal drugs against *Cn* and suggests the potential for developing new antifungal therapies based on entirely different principles applicable to a wide range of invasive fungal diseases.

## Introduction

Invasive fungal diseases (IFDs) pose a significant global health threat, particularly to immunocompromised populations. In 2022, the World Health Organization (WHO) published the first Fungal Priority Pathogens List (WHO FPPL), categorizing pathogenic fungi causing IFDs into three priority groups: critical, high, and medium [1,2]. The critical group includes four strains: *Cryptococcus neoformans* (*Cn*), *Candida albicans*, *Aspergillus fumigatus*, and *Candida auris*. *Cn* is a yeast-like airborne fungus with a global distribution found in environments such as soil, decaying wood, and pigeon droppings [3]. Unlike many other fungi and yeasts, *Cn* is thermotolerant and can thrive above 37°C, contributing to its virulence. Production of melanin pigment and thick capsules are also essential virulence factors for *Cn* [4]. Acquired

through the respiratory route, *Cn* primarily affects the lungs and can subsequently spread to the central nervous system in immunocompromised patients, such as those with HIV infection, leading to cryptococcal meningitis [3], which is responsible for 15% of AIDS-related death [5]. Existing anti-*Cn* agents target ergosterol synthesis (fluconazole), plasma membrane ergosterol (amphotericin B), and DNA/protein synthesis (flucytosine); however, these agents face challenges such as side effects and the emergence of resistant strains. Therefore, developing novel anti-*Cn* agents based on new molecular targets and innovative approaches is urgently needed [6].

This paper shows that the metabolism of the glycolipid, ergosterol β-glucoside (EG), in *Cn* is a new target for the development of anti-*Cn* agents. EG, a type of sterol β-glucoside (SG), is the major glycolipid of *Cn* with glucosylceramide (GlcCer). We have reported that EG is degraded in *Cn* by an enzyme that was found to be a homolog of endoglycocermidase (EGCase) and was named EGCase-related protein 2 (EGCrP2) [7]. Del Poeta *et al.* also independently named this enzyme sterylglucosidase 1 (Sgl1) [8]. The same gene, CNAG_05607 (*SGL1*), encodes EGCrP2 and Sgl1 (Gene ID: 23888906 in FungiDB). Although EGCase and EGCrP2/Sgl1 belong to the glycoside hydrolase (GH) family 5, the substrate specificity of the two enzymes is completely different [7,9,10]. This was also demonstrated by the X-ray crystal structure analysis of the complexes of EGCase with GM3 and EGCrP2 with EG (S1 Fig). The distribution of EGCase and EGCrP2/Sgl1 also differs significantly. EGCase is found in bacteria and some invertebrates [11] but not in fungi, whereas EGCrP2/Sgl1 is distributed explicitly within the fungal kingdom (S2 Fig). EGCrP2/Sgl1 is present in four of the most dangerous pathogenic fungi (*Cn, C. albicans, A. fumigatus,* and *C. auris*) listed in the WHO FPPL [1]. This indicates the potential to develop antifungal drugs against these IFDs by targeting EGCrP2/Sgl1. Neither EGCrP2/Sgl1 nor EGCase is found in mammals.

Deletion of the *SGL1* gene (CNAG_05607) [7,8] or use of inhibitors for EGCrP2/Sgl1 [10] leads to EG accumulation in *Cn* and attenuates the virulence in a mouse infection model. However, the molecular mechanism underlying the virulence attenuation in *SGL1*-deficient *Cn* (*sgl1Δ,* KO) has not been fully elucidated. Previously, EG was shown to accumulate in KO strain [8], mainly in its vacuoles [7]. This study extends these findings by revealing that EGCrP2/Sgl1 localizes predominantly to vacuoles and that EG accumulated in KO strain is converted to its acylated derivatives, acyl-EGs (AEGs).

Preliminary RNA sequencing (RNA-seq) data comparing KO strain with wild-type (WT) and *SGL1*-reconstituted *Cn* (*sgl1Δ::SGL1,* RE) strains suggested that a cluster of genes was strongly downregulated in KO strain at 37°C. Among these gene groups, we focused on autophagy, as the number of dead cells in KO strain significantly increased compared to WT and RE strains under nutrient starvation at 37°C. Autophagy (macroautophagy) is the process by which cellular components are trapped in organelles known as autophagosomes and transported to lysosomes/vacuoles for degradation and recycling under nutrient starvation [12]. In addition to its role in cell survival under nutrient starvation, autophagy has significant implications for physiological and pathological processes in eukaryotes. Autophagy-related genes (*ATG*s), extensively studied in budding yeast, are a group of conserved genes required for autophagy [13]. Recent studies indicate that several *ATG*s are essential for nutrient starvation adaptation and virulence in *Cn* [14,15].

Some microbial glycolipids are ligands for C-type lectin receptors (CLRs), and ligand binding to CLRs activates inflammatory cytokine production [16]. By screening CLRs responding to EG and AEGs using the Nuclear Factor of Activated T cells (NFAT)-GFP reporter cells expressing various CLRs, we found that Mincle interacts with EG and AEGs; the latter activated Mincle much stronger than the former. Mincle, a CLR mainly expressed in dendritic cells (DCs) and macrophages, is induced by various stimuli and triggers innate immune responses [17,18]. Ligand binding to Mincle triggers phosphorylation of the immunoreceptor tyrosine activation motif (ITAM) in the FcRγ chain, leading to NF-κB activation through an adaptor molecule called caspase recruitment domain family member 9 (CARD9). Structural analysis of Mincle shows that the carbohydrate recognition domain (CRD) has a unique structure not found in other CLR proteins [19]. In addition to two sugar-binding pockets, Mincle's CRD contains two hydrophobic pockets capable of binding fatty acids with more than ten carbon atoms. Glycolipids that interact with at least three of Mincle's four binding pockets can act as potent

ligands for Mincle [20]. In this paper, we discuss how mouse Mincle does or does not recognize the glycolipids accumulated in KO strain using binding models of mouse Mincle with either AEG or EG.

Overall, this study provides critical insights into the impact of vacuolar sterol β-glucosidase EGCrP2/Sgl1 deficiency on *Cn* pathogenicity and subsequent host immune responses and suggests that targeting EGCrP2/Sgl1 may pave the way for the rational design of a novel class of anti-cryptococcal drugs with broader implications for the treatment of IFDs.

## Results

### Loss of virulence of KO strain in a mouse infection model

KO strain exhibited reduced virulence in a mouse infection model compared with WT strain (Fig 1A). Similar results were obtained using a different strain of mouse [8]. RE strain was generated by expressing the *SGL1* gene in KO strain

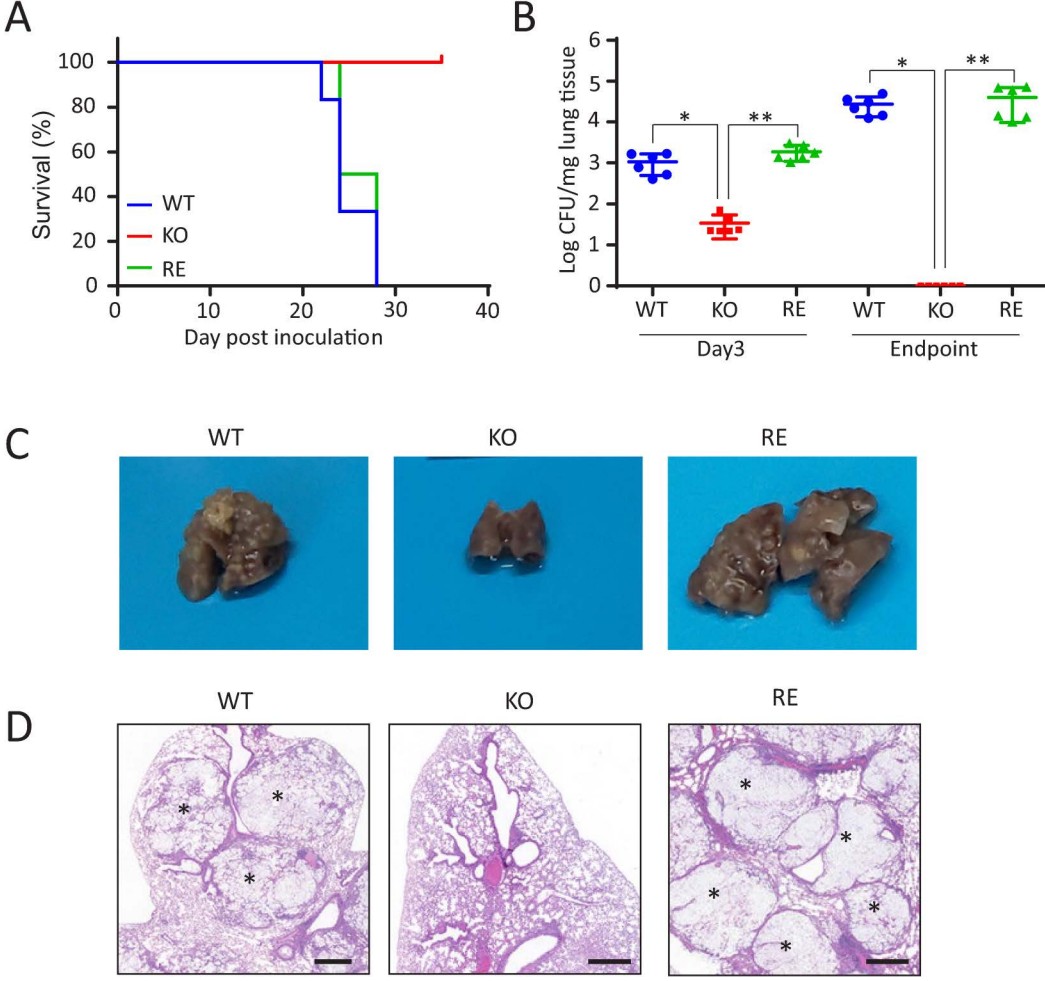

**Fig 1. Virulence of *C. neoformans* (*Cn*) in a mouse infection model.** A, Survival of mice infected with *Cn* WT, *sgl1Δ* (KO), and *sgl1Δ::SGL1* (RE) strains. Mice were intranasally administered $5 \times 10^5$ *Cn* cells. Each group consisted of 6 mice. B, Colony-forming units (CFU) in the lung on day 3 after infection and at the endpoint (WT, day 21-35; RE, day 22-28; KO, day 43). The values represent CFU per mg of lung tissue (n = 6). C, Appearance of the mouse lung at the endpoint after infection with WT, KO, and RE strains. D, Histology of mouse lung sections stained with hematoxylin and eosin at the endpoint after infection with WT, KO, and RE strains. Scale bar, 500 μm. Asterisks indicate cryptococcal lesions. Infection experiments were performed independently three times and data were reproducible.

(S3A-C Fig). The EGCrP2/Sgl1 activity in RE strain was restored to the level of WT strain (S3D Fig), and virulence in mice was restored in RE strain (Fig 1A). The number of colony-forming units (CFU) in the lungs of mice infected with KO strain was significantly reduced compared with those of WT and RE strains on day 3, and by the endpoint of the experiment, KO strain had nearly disappeared from the lungs (Fig 1B). Notably, mice infected with KO strain did not show lung swelling, as observed in mice with WT and RE strains (Fig 1C). Lung tissue sections from mice infected with WT and RE strains showed numerous lesions. In contrast, sections from mice infected with KO strain did not show such lesions (Fig 1D). These results indicate that EGCrP2/Sgl1 is essential for *Cn* virulence in a mouse infection model.

## Growth inhibition of KO strain

To investigate the cause of the loss of virulence in KO strain in a mouse infection model, we examined the effects of EGCrP2/Sgl1 deletion on the growth characteristics of *Cn* at different growth temperatures. Unlike previous studies in which *Cn* was cultured under hypoxic conditions [10], in this study, *Cn* was cultured in YPD medium with sufficient oxygen supply. Disruption of the *SGL1* gene had almost no effects on *Cn* growth until day 2 (log phase) at 25, 30, and 37°C when growth was estimated by measuring medium turbidity (OD at 600 nm); however, the growth of KO strain was reduced at all temperatures after day 3 (stationary phase) when nutrients in the medium would be depleted (Fig 2A). The decrease in turbidity (OD at 600 nm) after day 3 at 37°C was slower than that observed at 25 and 30°C. This may be due to leakage of cell contents derived from ruptured cells into the medium at 37°C.

Cell viability was examined by staining with phloxine B, a well-validated reagent for staining dead cells in budding yeast research [21,22]. In our control experiment, all *Cn* cells were fully stained with phloxine B after heat treatment (S4A Fig), and no colonies formed on the YPD agar medium, regardless of the presence or absence of the *SGL1* gene (S4B Fig).

We found that the number of phloxine B-positive cells (dead cells) significantly increased in KO strain when cultured at 37°C for 3 days compared with WT and RE strains (Fig 2B-D). The increase in dead cells in KO strain was temperature-dependent: the effect was observed at 37°C on day 3, 30°C on day 4, and 25°C on day 7 (S5 Fig). These results indicate that the loss of EGCrP2/Sgl1 increases the mortality rate of *Cn*, leading to a more rapid onset of cell death when KO strain is incubated at 37°C.

## Gene downregulation in KO strain at 37°C

To elucidate the molecular mechanism behind the increased mortality of KO strain at 37°C, we performed RNA-seq to identify genes with significantly altered expression in KO strain at 37°C. The preliminary RNA-seq data (a single replicate) suggests that the gene expression patterns were classified into four clusters by comparing the gene expression patterns of KO strain cultured at 30 and 37°C for 3 days with those of WT and RE strains (S6A Fig): a group in which gene expression is lower at 37°C compared with 30°C in all WT, KO, and RE strains (cluster A); a group in which gene expression is increased at 37°C compared with 30°C in WT and RE strains but not altered in KO strain (cluster B); KO strain gene expression is upregulated compared with WT and RE strains at both 30 and 37°C (cluster C); and gene expression in WT, KO, and RE strains is upregulated at 37°C compared with 30°C (cluster D). We focused on cluster B, where gene expression was significantly reduced in KO strain compared with WT and RE strains at 37°C because, on day 3, KO strain is characterized by a significant increase in the number of dead cells at 37°C compared with WT and RE strains. Cluster B is enriched with genes associated with membrane proteins, zinc-finger proteins, and autophagy (S6B Fig). Autophagy has been reported to enable fungi to adapt to nutrient limitations by recycling cellular components and is involved in the pathogenicity of pathogenic fungi [23]. We then examined the expression of *ATG* genes in KO strain cultured at 37°C for 3 days by qPCR using *ACT1* and *GAPDH* as reference genes. It was elucidated that *ATG* genes of KO strain were downregulated compared with WT and RE strains (Fig 3), indicating that the autophagy of KO strain is dysfunctional at the expression level of *ATG* genes when cultured at 37°C for 3 days.

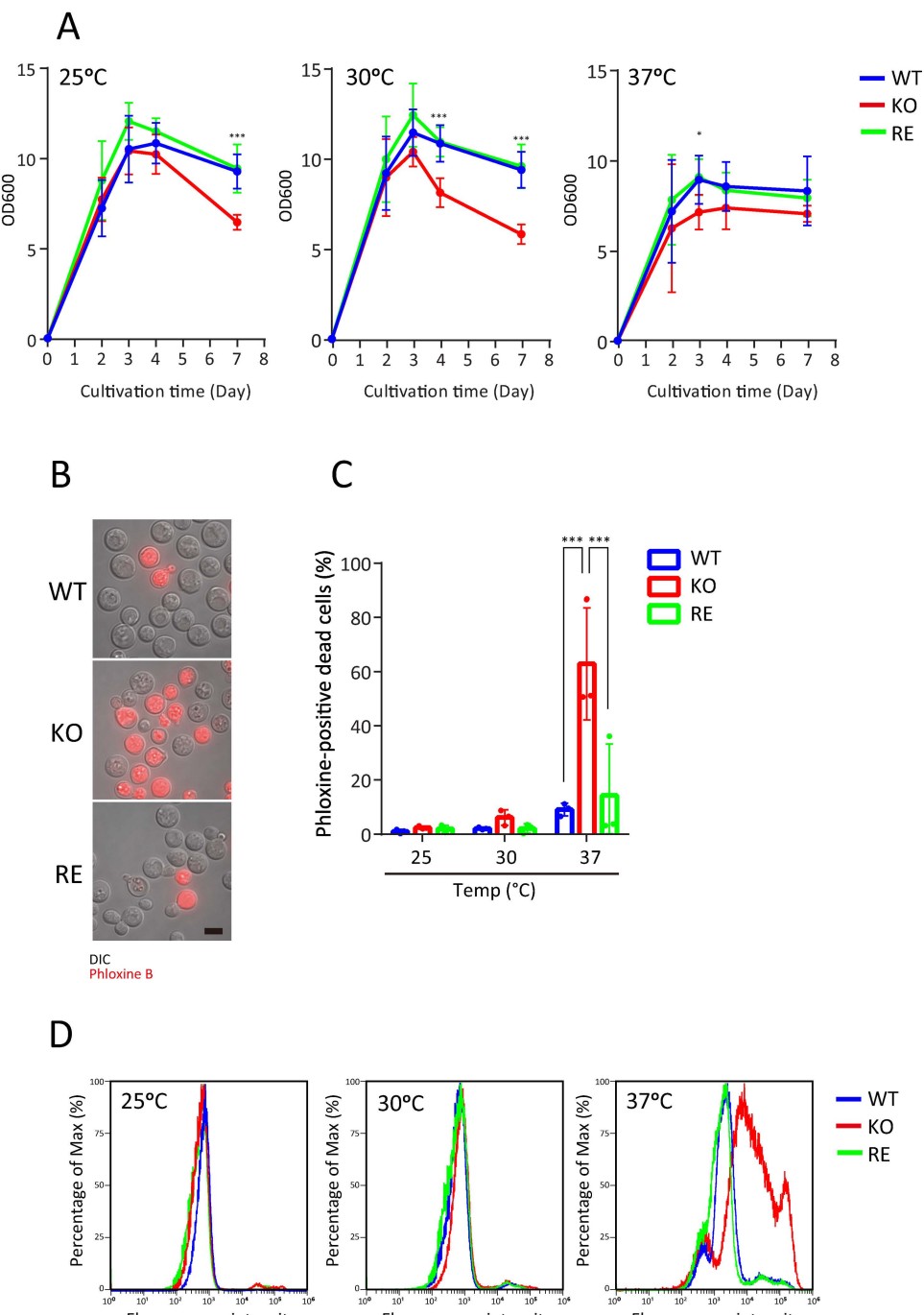

**Fig 2. Growth inhibition in KO strain at 37°C.** A, Cell growth curves of *Cn* at different temperatures. WT, KO, and RE strain were cultured in YPD medium at 25, 30, and 37°C with shaking at 150 rpm. Optical density at 600 nm ($OD_{600}$) was measured by a spectrophotometer on the indicated days during cultivation. Each experiment was performed with 5-7 replicates (25°C WT, *n* = 7; 25°C KO, *n* = 5; 25°C RE, *n* = 5; 30°C WT, *n* = 7; 30°C KO, *n* = 6; 30°C RE, *n* = 6; 37°C WT, *n* = 7; 37°C KO, *n* = 5; 37°C RE, *n* = 5). B, Phloxine B staining of *Cn* cells. WT (upper), KO (middle), and RE (lower). *Cn* were cultured in YPD medium for 3 days at 37°C with shaking at 150 rpm. Dead cells (red) were stained with phloxine B and captured by a fluorescence microscope [21]. Scale bar, 5 μm. C, Quantification of phloxine B positive cells. Numbers of phloxine B-positive cells were counted under a fluorescence microscope. Each experiment analyzed at least 100 cells (n = 3). D, Flow cytometry of *Cn* cells stained with phloxine B. WT, KO, and RE strain were cultured in YPD medium for 3 days at 25, 30, and 37°C with shaking at 150 rpm. Cells were stained with phloxine B and analyzed by a flow cytometer using a BL2-H channel (574/26 nm BP filter).

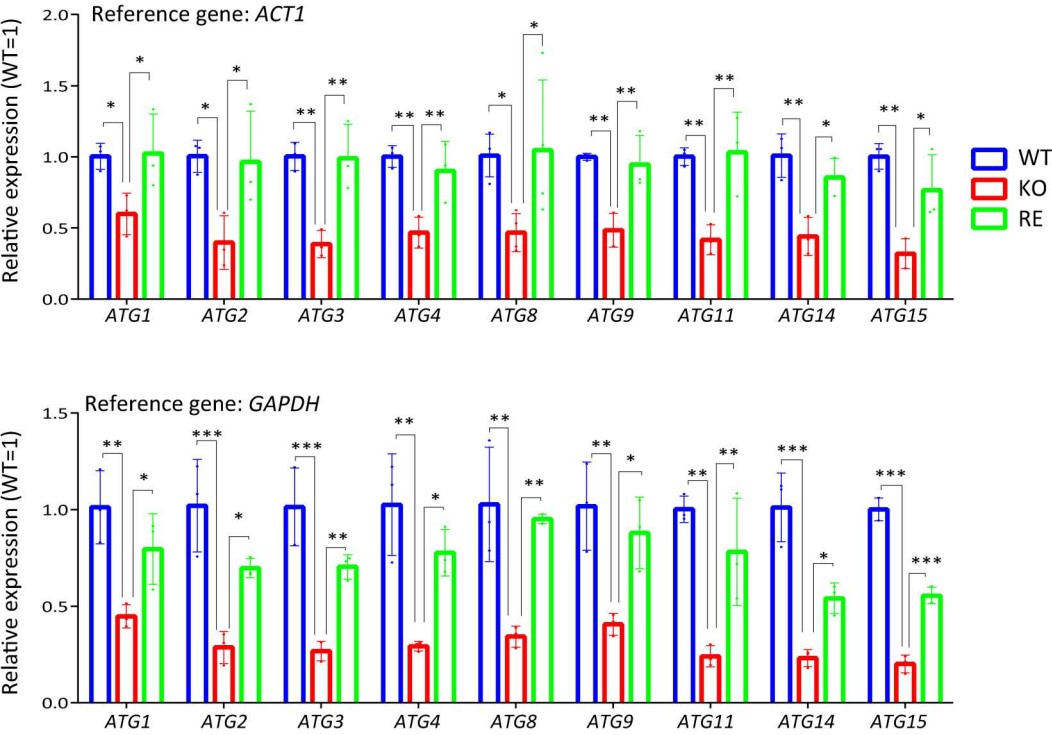

**Fig 3. The genes downregulated in KO strain at 37°C.** Relative mRNA expression levels were determined using *ACT1* and *GAPDH* as reference genes with PCR amplification efficiency collection. Data are presented as Mean ± SD (n = 3).

## Autophagy dysfunction in KO strain

To investigate the possibility of dysfunction in autophagy in KO strain, we monitored the progression of autophagy by tracking GFP-Atg8 expressed in WT, RE, and KO strain. GFP-Atg8 binds to phosphatidylethanolamine (PE) on autophagosome membranes, and after fusion, it is translocated into the vacuole, where proteases rapidly degrade Atg8. In contrast, GFP remains in the vacuole due to its resistance to protease degradation. Thus, GFP signals can be seen within vacuoles if autophagy proceeds normally [24]. First, we investigated the subcellular localization of GFP-Atg8 in WT, KO, and RE strains expressing GFP-Atg8 cultured in YPD medium at 37°C. On day 3, GFP signals were observed to translocate into the vacuoles of WT and RE strains but not KO strain (Fig 4A and 4B). Vacuolar localization of GFP in WT and RE strains was further confirmed by FM4–64 staining (S7 Fig). Consistent with these results, the proportion of dead cells was significantly higher in KO strain compared with WT and RE strains on day 3 (Fig 2B-D). These findings suggest that autophagy is successfully induced in WT and RE strains but fails in KO strain as nutrients become depleted on day 3.

To further confirm autophagy induction, we conducted experiments using nitrogen-free medium (SD-N medium) at 37°C instead of YPD medium. In WT and RE strains, GFP signals were detected in vacuoles as early as 1 hour after transfer to SD-N medium, and vacuolar localization became more prominent over time, with approximately 80% of GFP signals localized to vacuoles after 4 hours of incubation. In contrast, no vacuolar localization of GFP signals was observed in KO strain, even after 4 hours of incubation (Fig 5A and 5B).

The degradation of GFP-Atg8 in *Cn* cells cultured in SD-N medium was analyzed by Western blot using an anti-GFP antibody. In WT and RE strains, GFP-Atg8 degradation and the accumulation of free GFP increased progressively over

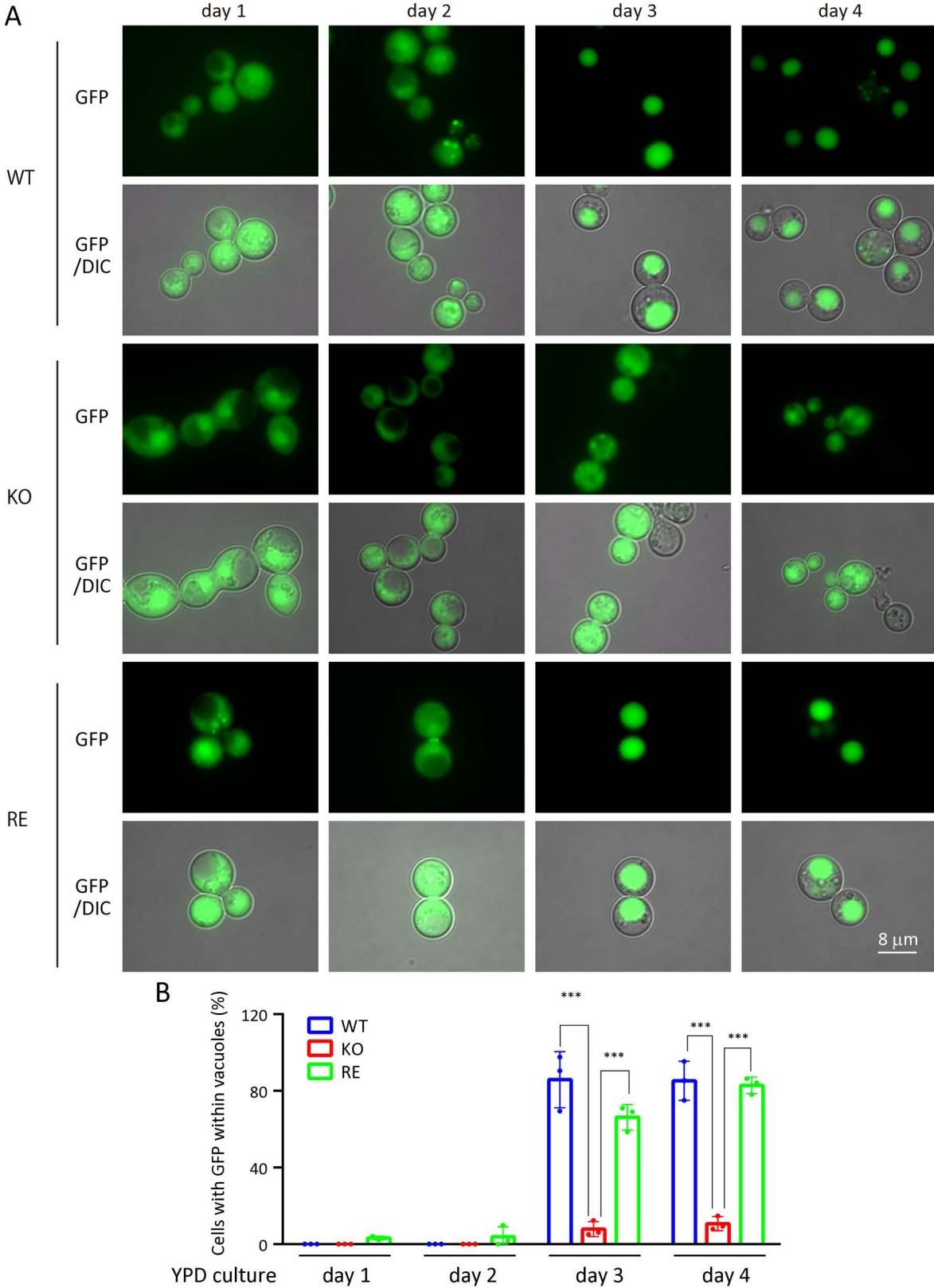

**Fig 4. Autophagy progression in WT, KO, and RE strains at 37°C in YPD medium.** A, Localization of GFP-Atg8 in WT, KO, and RE strains cultured at 37°C for different periods (from day 1 to day 4). Fluorescent images of GFP were obtained using the FITC filter set and merged with

differential interference contrast (DIC) images. B, Ratio of the GFP signal within the vacuole. Numbers of cells showing GFP within the vacuole were counted under a fluorescence microscope. The percentage of cells with GFP within vacuoles was calculated as: Cells with GFP within vacuoles (%) = cells with GFP within vacuoles/ cells with GFP within vacuoles + cells with GFP but not in vacuoles x 100. Each experiment analyzed at least 100 cells (n = 3).

time. In contrast, KO strain exhibited significantly lower levels of GFP-Atg8 degradation and free GFP production compared with WT and RE strains (Fig 6). These results confirm that KO strain fails to GFP-Atg8 translocation to vacuoles.

We found that prolonged incubation in SD-N medium for over 4 hours resulted in a significantly higher proportion of dead cells in KO strain than WT and RE strains (Fig 7A and 7B). Notably, no increase in dead cell numbers was observed in KO strain cultured in YPD medium for 8 hours. These findings indicate that autophagy dysfunction is the primary cause of the increased cell death observed in KO strain under these conditions.

Collectively, these results demonstrate that KO strain fails to induce autophagy under nutrient starvation at 37°C, which ultimately leads to cell death.

## Intracellular localization of EGCrP2/Sgl1 in *Cn*

In *Cn*, disruption of the *SGL1* gene (CNAG_05607) leads to the accumulation of EG, suggesting that the intracellular substrate for EGCrP2/Sgl1 is EG [7,10]. Although EG is accumulated in vacuoles in KO strain [7], the intracellular localization of EGCrP2/Sgl1 in *Cn* has yet to be clarified. In this study, we investigated this using mRuby-labeled EGCrP2/Sgl1. As a result, we found that mRuby-labeled EGCrP2/Sgl1 was predominantly localized to vacuoles, as indicated by quinacrine staining (Figs 8A and S8). In contrast, signals were observed throughout the cytosol when free mRuby was expressed in *Cn* (Figs 8A and S8). The acidic pH optimum of *Cn* EGCrP2/Sgl1 suggests that it functions in vacuoles (Fig 8B). These results indicate that EGCrP2/Sgl1 localizes in vacuoles where it is responsible for the catabolism of EG, despite CNAG_05607 being categorized as a cytoplasmic protein in FungiDB (https://fungidb.org/fungidb/app/record/gene/CNAG_05607).

## Accumulation of EG and AEGs in KO strain

**Identification of AEGs accumulated in KO strain.** KO strain showed a marked accumulation of EG, which was hardly detectable in WT and RE strains by TLC (Fig 9A). The presence or absence of EGCrP2/Sgl1 did not affect the level of GlcCer, another major glycolipid in *Cn,* indicating that the endogenous substrate of EGCrP2/Sgl1 in *Cn* is EG. An unknown lipid (tentatively named lipid Y) with a higher Rf value on thin-layer chromatography (TLC) accumulated in KO strain but not WT and RE strains (Fig 9A). Upon purification (Fig 9B and 9C) and LC-ESI MS/MS analysis (Fig 9D), lipid Y was identified as acylated EG (AEG). In LC-ESI MS/MS, the precursor ions $m/z = 838.7$ and $m/z = 840.6$ were attributed to AEG containing 18:2 and 18:1 fatty acids, respectively. The ergosterol-derived fragment ion $m/z = 379$ was commonly generated from precursor ions $m/z = 838.7$ (18:2-EG) and $m/z = 840.6$ (18:1-EG) with low collision energy (CE). Using high CE, fragment ions 145 and 159 were explicitly generated from double bonds at Δ5 and Δ7 of the B-ring in ergosterol, respectively (Fig 9D)[25]. Gas chromatography analysis revealed that AEG contained 16:0, 18:0, 18:1n-9, or 18:2n-6 (Fig 9E), with 18:1-EG being the most abundant. AEG progressively increased during the culture of KO strain but was hardly detectable in WT strain after prolonged incubation (Fig 9F).

**Effect of temperature on accumulation of EG and AEGs in KO strain.** We examined the effect of culture temperature on the accumulation of EG and AEGs in KO strain on days 2 and 3. The amount of EG accumulated in KO strain remained consistent across all examined temperatures (Fig 10A). In contrast, the accumulation of AEGs was significantly higher at 37°C compared with 25 and 30°C in KO strain (Fig 10B). Notably, the accumulation of EG and AEGs was barely detected in WT and RE strains at any temperature (Fig 10A and 10B).

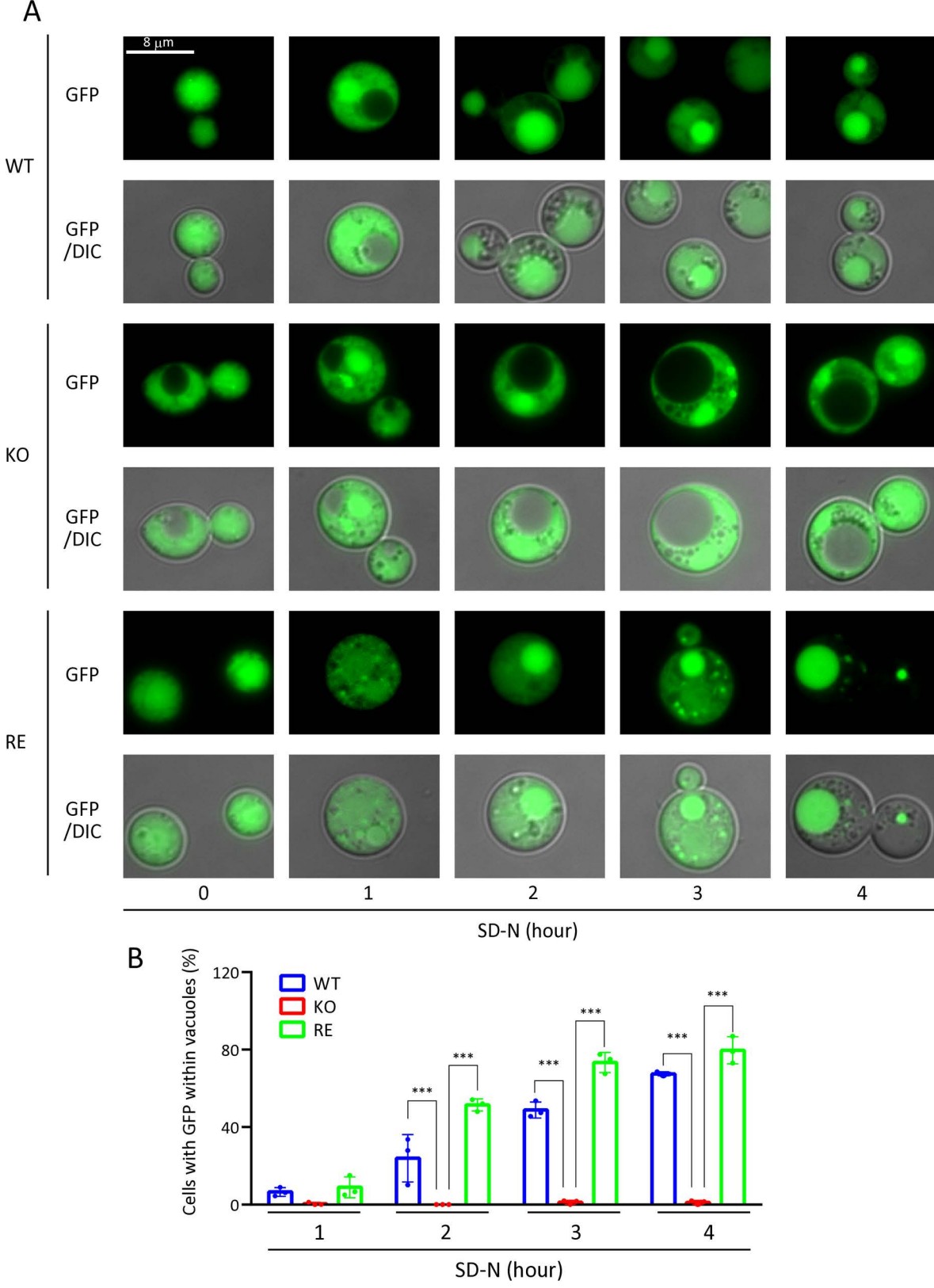

**Fig 5. Autophagy progression in WT, KO, and RE strains at 37°C in SD-N medium.** A, Autophagy induction by translocation of GFP-Atg8 in WT, KO, and RE strains vacuoles after transfer to SD-N medium. Cn was cultured in YPD medium for 15 hours at 30°C, then collected by centrifugation (5,000×g for 3 minutes). Cn cells were washed three times with PBS, transferred to SD-N medium, and incubated at 37°C for the indicated periods. GFP

fluorescence images were captured using a fluorescence microscope with the FITC filter and merged with differential interference contrast (DIC) images. B, Ratio of the GFP signal within the vacuole. The number of cells showing GFP within the vacuole was counted under a fluorescence microscope. The percentage of cells with GFP within vacuoles was calculated as: Cells with GFP within vacuoles (%) = cells with GFP within vacuoles/ cells with GFP within vacuoles+cells with GFP but not in vacuoles×100. Each experiment analyzed at least 100 cells (n=3).

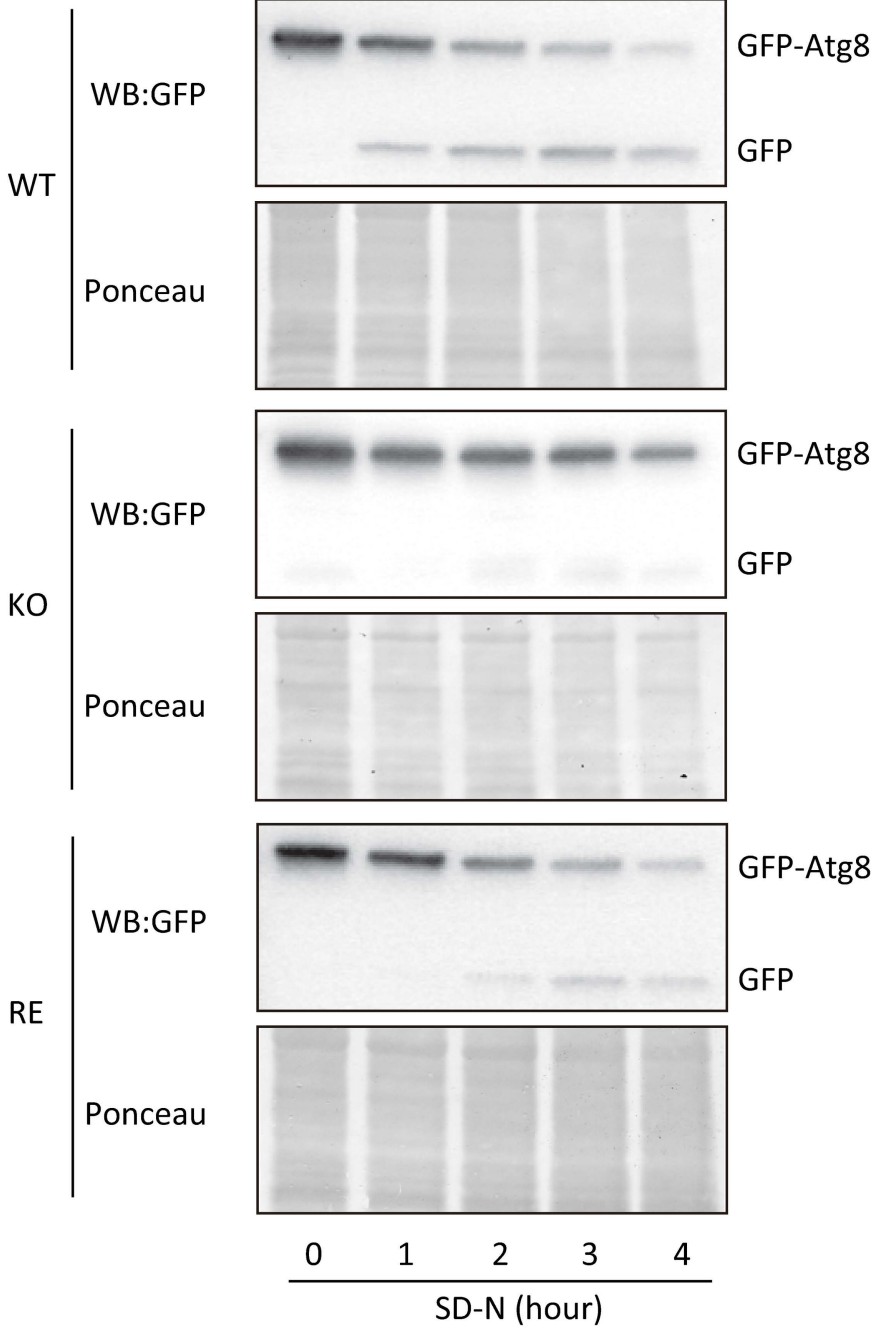

**Fig 6. Autophagy progression in WT, KO, and RE strains at 37°C evaluated by cleaving GFP-Atg8.** Immunoblot analysis using an anti-GFP antibody evaluated the autophagy progression. During the normal autophagy process, GFP-Atg8 (ca. 42 kDa) is cleaved in vacuoles, generating free GFP (ca. 27 kDa) [24]. *Cn* cells were transferred to an SD-N medium and incubated at 37°C for the indicated periods. Experimental details are described in Materials and methods. The experiments were conducted twice, and the data were reproducible. Ponceau shows the proteins subjected to Western blotting before being treated with anti-GFP antibodies.

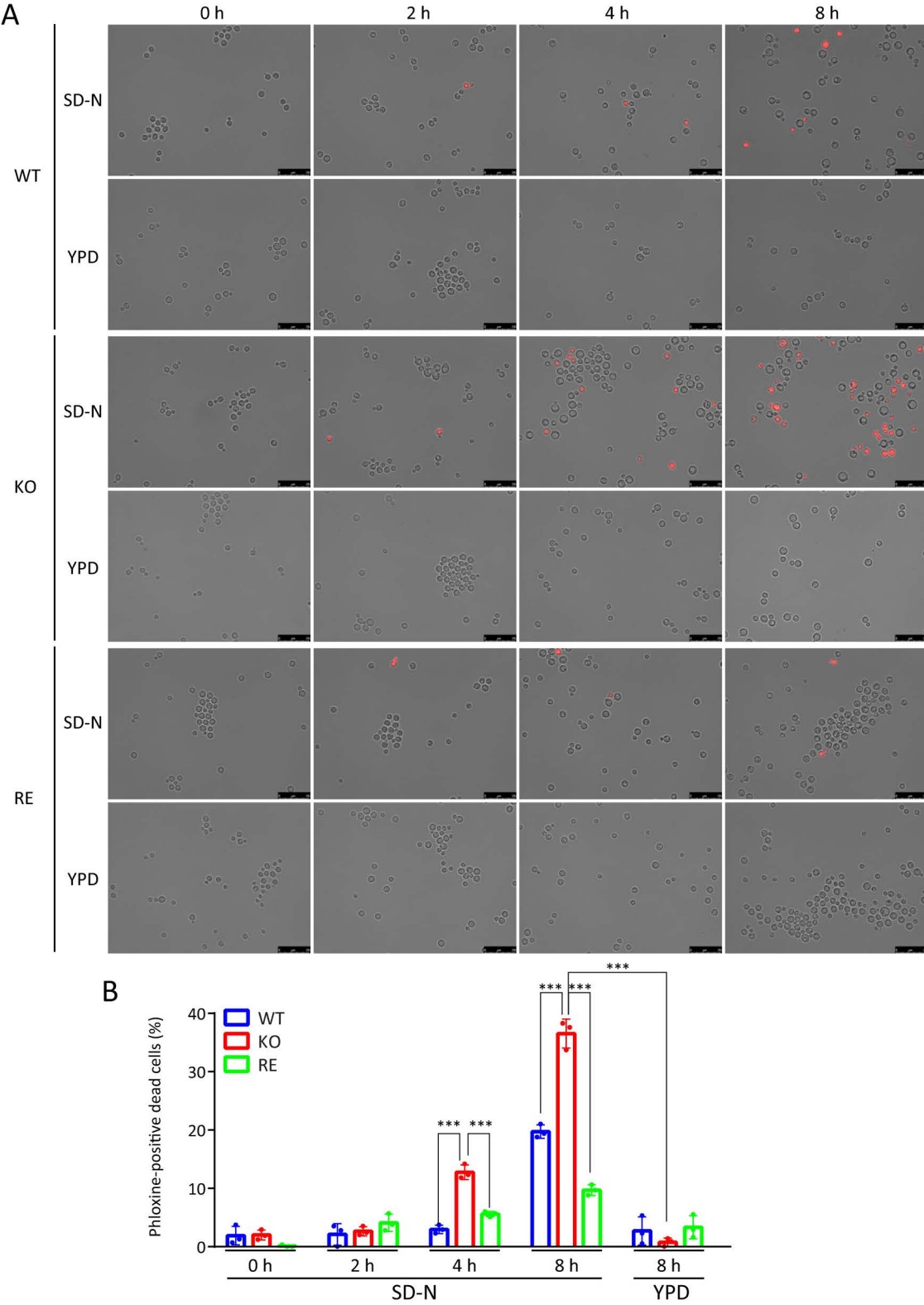

**Fig 7. The impact of autophagy deficiency on the survival of the KO strain under nitrogen starvation at 37°C.** A. Detection of dead cells of WT, KO, and RE strains cells cultured under nitrogen-starved condition (SD-N) at 37°C. Dead cells were stained with phloxine B and observed under a

fluorescence microscope at the time points indicated in the picture. Each strain cultured in nutrient rich YPD medium was used as a control. Scale bar, 25 μm. B, Quantification of phloxine B-positive cells. Numbers of phloxine B-positive cells were counted under a fluorescence microscope. Each experiment analyzed at least 150 cells (n = 3).

**Accumulation of SGs and ASGs in *A. fumigatus*.** The WHO classifies *A. fumigatus* as a particularly threatening pathogenic fungus similar to *Cn* [1]. An EGCrP2/Sgl1 homolog of *A. fumigatus* was identified (SglA, AFUA_3G00820), and disruption of AFUA_3G00820 resulted in the accumulation of various SGs, including EG, in *A. fumigatus*. Interestingly, the SglA-deficient *A. fumigatus* (*sglA*Δ, KO) was avirulent in a mouse infection model [26]. We investigated whether acylated SGs (ASGs) accumulate in the KO strain when cultured in YPD medium at 37°C. The strategies for disrupting AFUA_3G00820 and PCR of WT, KO and *sglA*Δ::*SGLA* (RE) strains are shown in S9A-C Fig. As a result, we found that EG and other SGs were accumulated in conidia (S9D Fig) and hypha (S9F Fig) of KO strain, as reported in [26]; in addition, AEGs were also accumulated in conidia (S9E Fig) and hyphae (S9G Fig) of KO but not WT or RE strains. *Cn* has EG as the sole SG, but *A. fumigatus* has various SGs besides EG. As a result, various ASG molecules were produced in the conidia and hypha of KO strain, with AEGs being the most abundant. Notably, the ratio of AEGs to EG in KO strains is markedly higher in *A. fumigatus* than *Cn* under the same culture conditions. These results suggest that in fungi possessing *SGL1* gene, deletion of this gene leads to the accumulation of SG and ASGs at varying molar ratios, depending on the fungal species.

## The activation of CLR Mincle by SGs and ASGs

Some microbial glycolipids are ligands for CLRs, activating the innate immune system [16]. Therefore, we investigated whether EG and AEGs, both accumulated in KO strains of *Cn* and *A. fumigatus*, act as ligands for CLRs. We used the NFAT-GFP reporter cells expressing various CLRs (Fig 11A). Our results showed that EG and AEGs specifically activated Mincle but not other CLRs under the conditions used (Fig 11B). To further validate the specificity of Mincle activation by SGs and ASGs, we quantitatively examined the activities of SGs containing different sterol species as aglycon moieties using alkaline phosphatase (SEAP) reporter 293 cells expressing mouse Mincle (Fig 11C). We found that SGs containing plant sterols (stigmasterol and sitosterol) or cholesterol did not activate mouse Mincle even at high concentrations (10 nmol/well). In contrast, ASGs including acylated sitosterol β-glucoside (Acyl-sitoSG, ASiG) strongly activated mouse Mincle, even at lower concentrations (0.05-0.1 nmol/well) (Fig 11D). The degree of activation of mouse Mincle by ASGs was similar to that of trehalose-6,6-dibehenate (TDB), a known potent ligand for Mincle, under the conditions used (Fig 11E). We chemically synthesized an EG and two AEGs (16:0-EG and 18:1-EG) (S10 Fig) and used them for assays of Mincle activation using NFAT-GFP reporter cells expressing Mincle from different origins. The activation of Mincle by EG depended on the origins of Mincle, *i.e.,* chemically synthesized EG (1 nmol/well) activated human and rat Mincle, but it did not activate mouse Mincle even at higher concentrations (Fig 11F). Similarly, glycerol monomycolate has been reported to activate human but not mouse Mincle [27]. In contrast, chemically synthesized AEGs activated all Mincle tested regardless of origins, *i.e.*, 16:0-EG (Fig 11G) and 18:1-EG (Fig 11H) showed potent activation of mouse, rat, and human Mincle. The degree of Mincle activation by AEGs was comparable to that by TDB (Fig 11I) and was 10–50 times more potent than EG in activating human and rat Mincle (Fig 11F-H). On the other hand, molecular species of sterol in ASGs did not influence the degree of Mincle activation (Fig 11D, 11G and 11H). Notably, cells expressing FcRγ but not Mincle (control) did not react with EG, AEGs, or TDB (Fig 11F-I).

## Analysis of Mincle binding models with either AEG or EG

**Significance of Ca$^{2+}$ ion for binding of Mincle with AEG and EG.** To better understand why mouse Mincle is strongly activated by AEG but not EG, we constructed docking models of mouse Mincle with either AEG (18:1-EG) or EG. The binding model suggests important roles for Ca$^{2+}$ in the interaction of mouse Mincle with these glycolipids (calcium ions are shown as yellow spheres, Fig 12). The EPN motif (residues 169–171 in mouse Mincle) may contribute to glucose

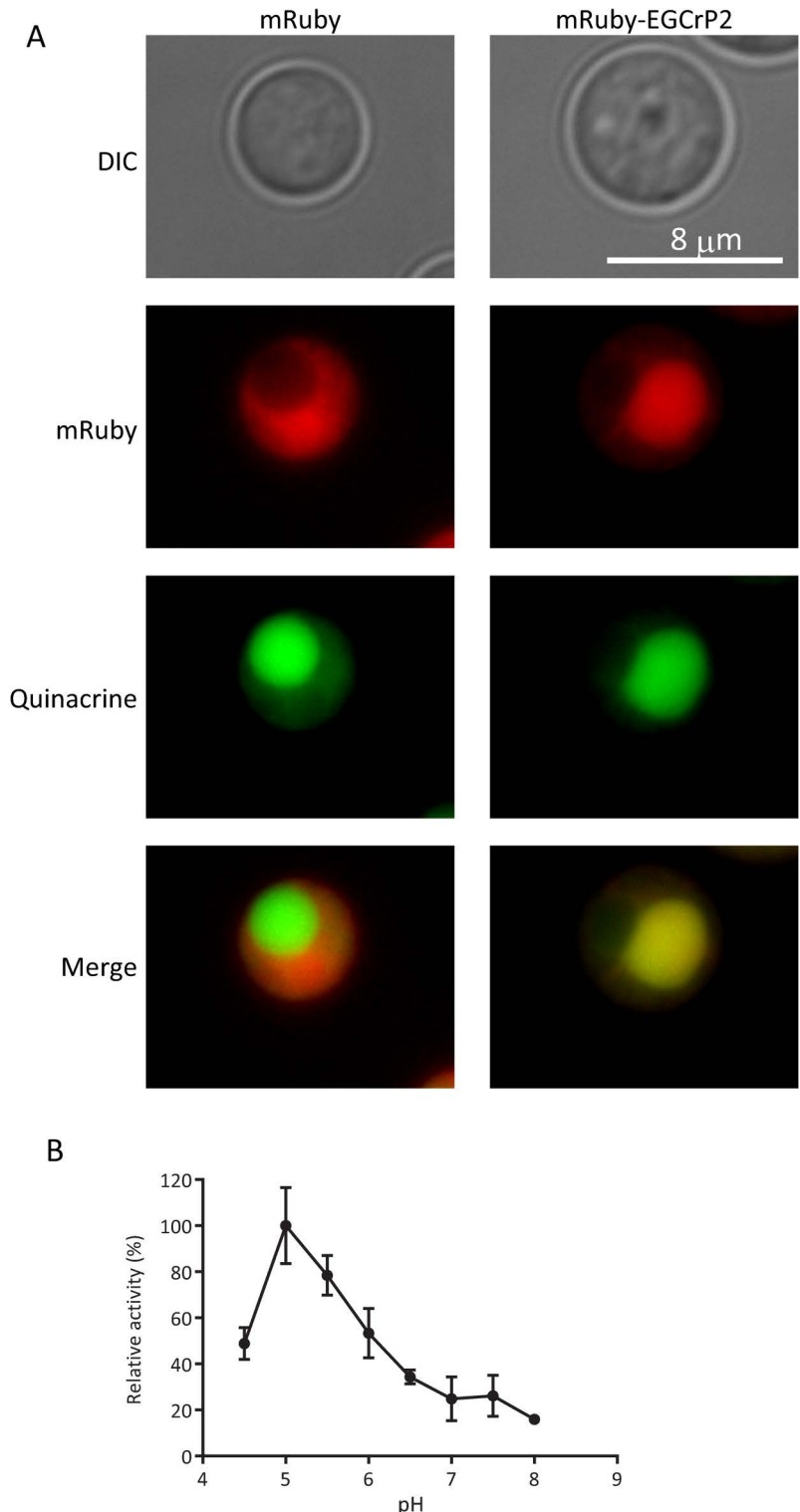

**Fig 8. Subcellular localization of EGCrP2/Sgl1 in *Cn* and its optimum pH.** A, Subcellular localization of mRuby-EGCrP2/Sgl1 expressed in WT strain cultured at 25°C for 2 days. Fluorescence microscopy captured mRuby-derived fluorescence (red), showing EGCrP2/Sgl1 distribution (mRuby-EGCrP2/Sgl1) and free mRuby (mRuby). The vacuole was stained by quinacrine (green) that accumulates in fungal vacuoles [56]. Both signals were merged in mRuby-EGCrP2/Sgl1-expressing *Cn*, turning to yellow. B, Optimum pH of EGCrP2/Sgl1. The enzyme activity was measured using C6-NBD-GlcCer as a substrate and 150 mM GTA buffer varying pHs according to the method described in [7]. Data are presented as Mean ± SD (n = 3).

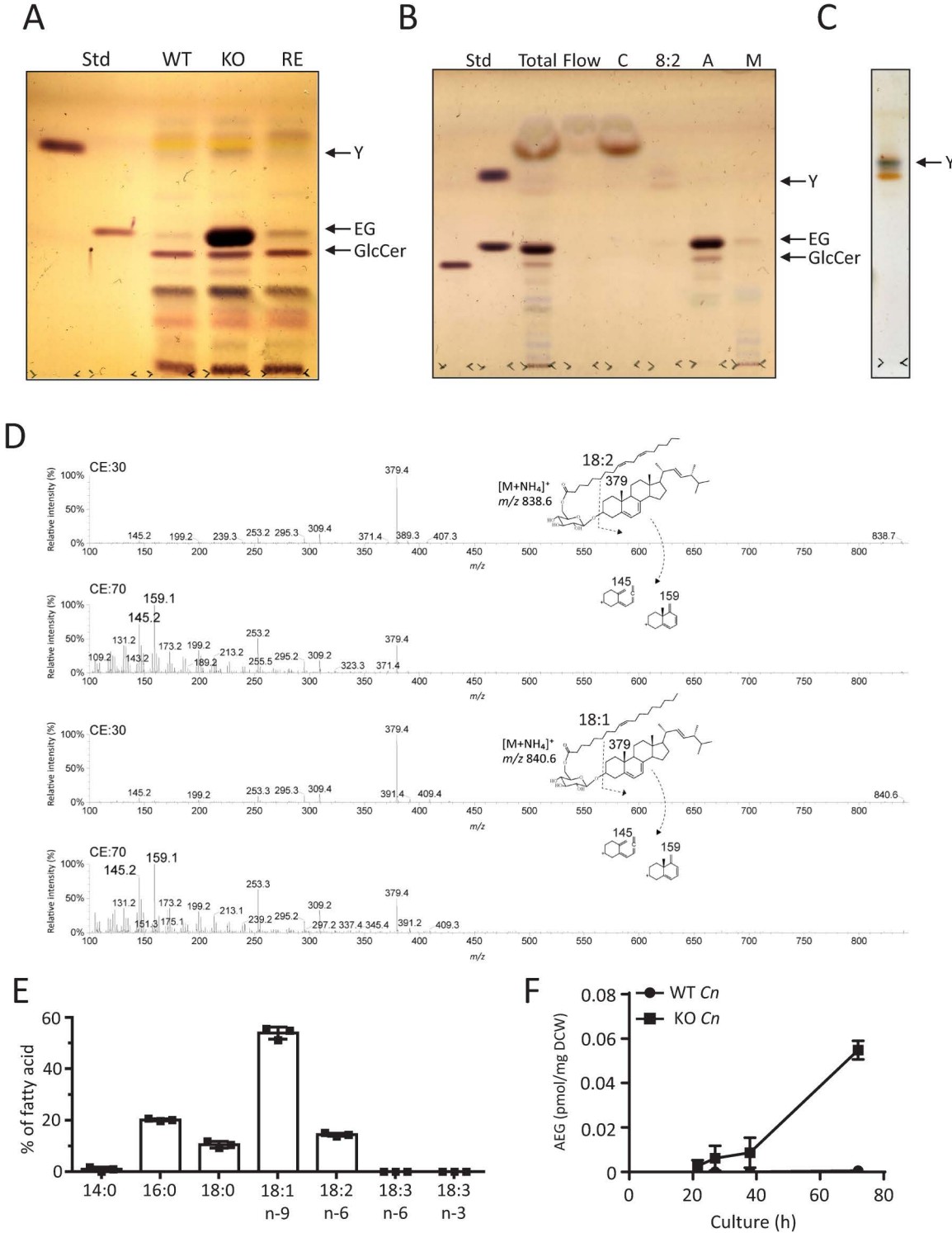

**Fig 9. Isolation and characterization of AEGs in KO strain.** A, TLC analysis of the glycolipids from WT, KO, and RE strains. Glycolipids were extracted from *Cn* (20 mg of dry cells), cultured in YPD medium for 3 days at 30°C, using chloroform/methanol (2/1, v/v). TLC was developed with chloroform/methanol/water (65/16/2, v/v/v) and stained with orcinol sulfate. Glycolipid Y shows an unidentified glycolipid accumulated in KO strain. B, C, Isolation of glycolipid Y from the total lipids of KO strain. Total lipids were extracted from KO strain using chloroform/methanol (2/1, v/v), and glycolipid Y was isolated using a Sep-Pak plus silica cartridge. Glycolipid Y was eluted from the cartridge with chloroform/acetone (8/2, v/v). D, Tandem mass analysis of glycolipid Y. The top and third spectra show the MS/MS fragmentation ions generated from precursor ions *m/z* 838 and 840, respectively, at

collision energy (CE) 30. The fragment ion *m/z* 379 was generated under this condition. Glycolipid Y was also applied to tandem mass using higher CE (CE70) to obtain more detailed structural information. The characteristic fragment ions *m/z* 145 and 159 derived from ergosterol were generated under this condition from precursor ions *m/z* 838 (second spectrum) and m/z 840 (bottom spectrum). The proposed structure of glycolipid Y (AEGs, 18:1-EG, and 18:2-EG) and its fragment pattern are shown above the spectrum. E, Fatty acid composition of AEGs. As described in Materials and methods, AEGs were subjected to gas chromatography after metanalysis. F, Time-course of AEG production in KO and WT strain. AEGs were extracted from *Cn* cells cultured at 30°C for the period indicated using chloroform/methanol (2/1, v/v) and quantified by LC-ESI MS/MS. Data are presented as Mean±SD (n=3). DCW, dry cell weight.

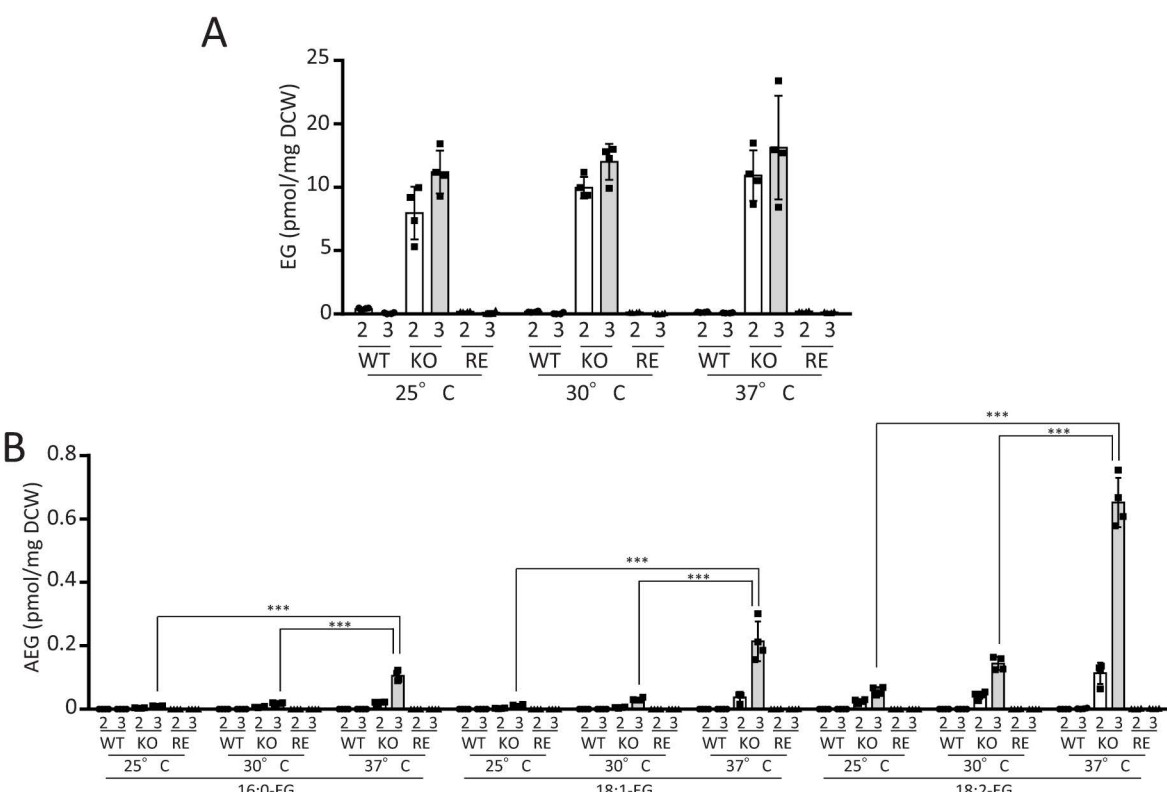

**Fig 10. Accumulation of EG and AEGs in KO strain at different temperatures.** A, Quantification of EG amount. B, Quantification of AEG amount. WT, KO, and RE strains were cultured at 25, 30, and 37°C for 2 and 3 days with shaking at 150 rpm. EG and AEG contents were quantified by LC-ESI MS/MS. Data are presented as Mean±SD (n=4). White and gray bars represent the amounts of EG and AEGs on day 2 and day 3, respectively. DCW, dry cell weight.

recognition via the $Ca^{2+}$ ion-mediated binding network at site 1, as has been reported for human Mincle [19,28]. It is predicted that the $Ca^{2+}$ ion at site 3, together with N172, D143, E147, and D178, forms a network, and the fatty acyl chain (18:1) of AEG binds to the hydrophobic region consisting of I195, P196, Y198, and Y199 (lipid-binding pocket), which is formed on this network. This suggests that the $Ca^{2+}$ ion at site 3 likely plays a role in stabilizing the interaction between AEG and Mincle (Fig 12A). In contrast, EG lacks this fatty acyl chain, and therefore, the $Ca^{2+}$ ion at site 3 may not contribute to EG binding to mouse Mincle (Fig 12B).

**Sugar and lipid-binding pockets in mouse Mincle.** Crystal structure analysis revealed that Mincle has two sugar-binding and two lipid-binding pockets and requires interaction with three of these four binding pockets to function as

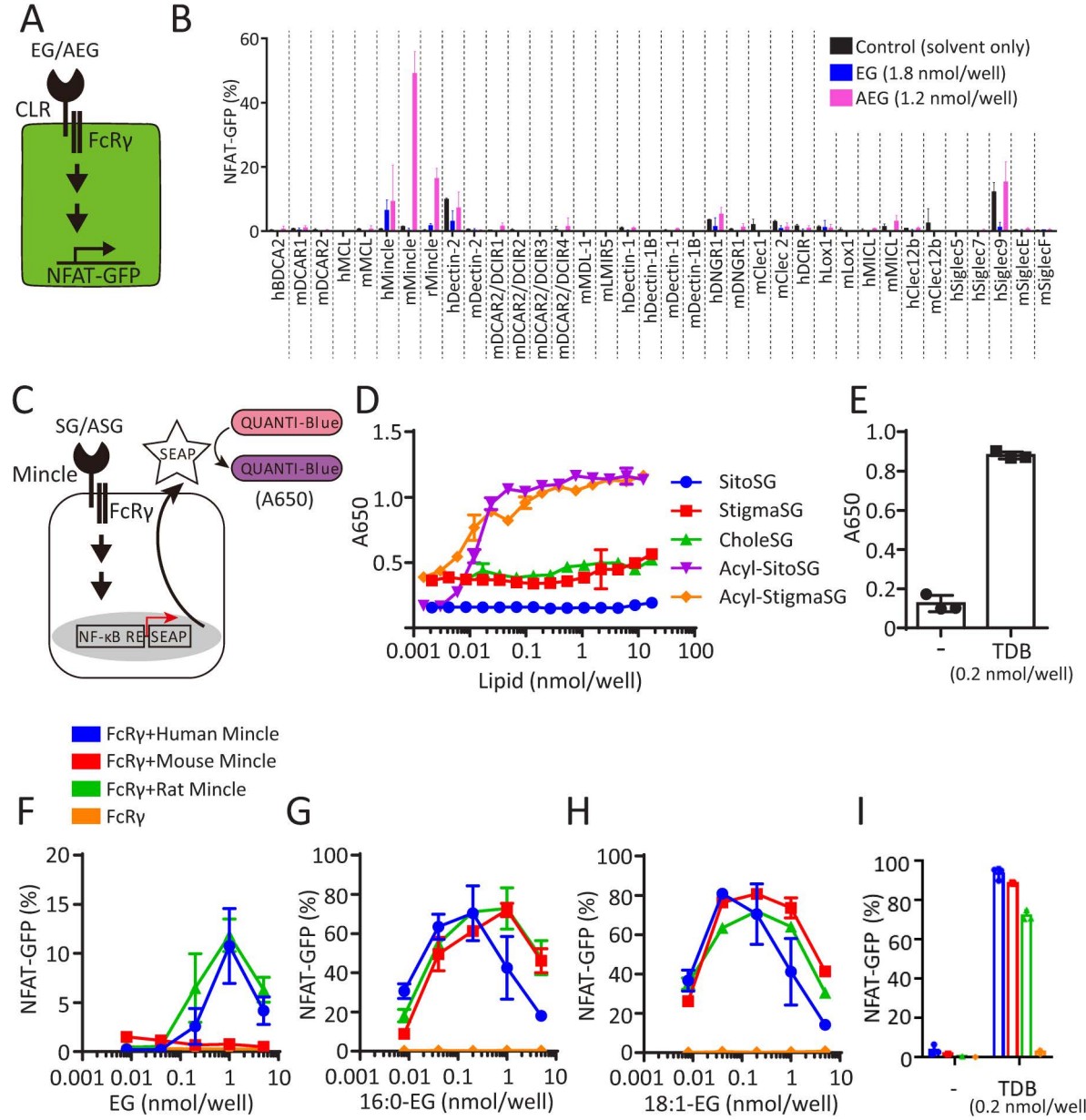

**Fig 11. Activation of Mincle by SGs and ASGs.** A, Experimental design for measuring SG and ASG activities on CLRs using NFAT-GFP reporter cells expressing CLRs. B, Activities of EG and AEGs on various CLRs. Reporter cells expressing CLRs were exposed to glycolipids (EG, 1.8 nmol/well; AEGs, 1.2 nmol/well) or 2-propanol (IPA, negative control) for 18 h on a 96-well plate. Flow cytometry was employed to analyze GFP expression in reporter cells. The percentage of NFAT-GFP was calculated as follows: NFAT-GFP (%) = GFP-positive cells/ total cells x 100. C, Experimental design for measuring SG and ASG activities for Mincle using secreted alkaline phosphatase (SEAP) reporter in HEK293 cells expressing Mincle. Activation of NF-κB was induced by Mincle stimulation, and secreted SEAP was quantified using QUANTI-Blue as a substrate. D, E, Activation of Mincle by SG, ASGs, and TDB (authentic Mincle ligand, 0.2 nmol/well). The activity was measured by SEAP reporter assay. -, IPA without TDB. SitoSG, sitosterol β-glucoside; StigmaSG, stigmasterol β-glucoside; CholeSG, cholesterol β-glucoside; Acyl-SitoSG, acylated sitosterol β-glucoside (ASiGs); Acyl-StigmaSG, acylated stigmasterol β-glucoside. F-I, Activation of various Mincle by EG, AEGs (16:0-EG and 18:1-EG), and TDB (0.2 nmol/well). The activity was measured by NFT-GFP reporter assay. EG and AEGs were chemically synthesized, and their structures are presented in S7 Fig. All data are presented as Mean ± SD (n = 3).

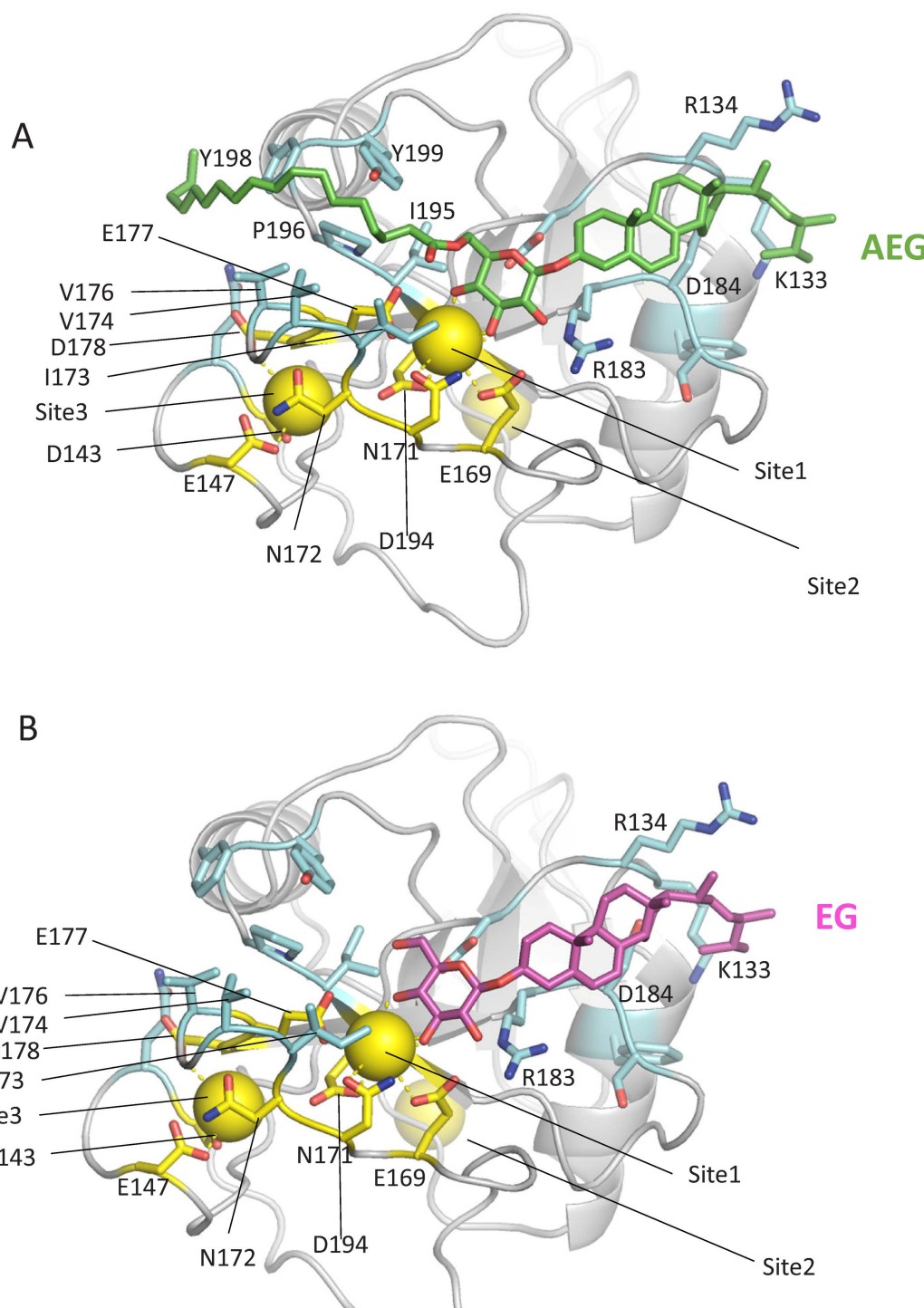

**Fig 12. Significance of Ca²⁺ ion for binding of Mincle with AEG and EG.** A, AEG docking model (18:1-EG, colored in green); B, EG docking model (EG, colored in purple). Residues involved in calcium binding are highlighted in yellow. Residues directly interacting with AEG and EG are shown in blue. Calcium ions are represented as yellow spheres. We superimposed the mouse Mincle on the bovine Mincle structure (PDB ID: 5KTH) after truncating the first 62 N-terminal residues. Guided by the three Ca²⁺-binding sites in the bovine Mincle [28], a Ca²⁺-bound conformation for the mouse Mincle was proposed.

a potent ligand [19,20]. The docking models of mouse Mincle with EG and AEG (S11 Fig) suggest that glucose and ergosterol of EG interact with the EPN motif (sugar-binding pocket) and the lipid-binding pocket (hydrophobic region shown in white in S11A and S11C Fig), respectively. In addition, the fatty acyl chain of AEG interacts with an additional lipid-binding pocket via the site 3 $Ca^{2+}$ network, as shown above. These results suggest that AEGs fulfill the ligand binding requirements of mouse Mincle more effectively than EG, which may explain why AEGs activated Mincle more strongly than EG in the experiments.

## Extracellular transport of EG and AEGs

**Transport of EG and AEGs by extracellular vesicles (EVs).** Mincle is primarily expressed on the plasma membrane of DCs and macrophages [18]. To understand how EG and AEGs activate Mincle *in vivo*, we investigated the possibility of their extracellular transport from *Cn* via EVs (Fig 13A). When *Cn* is cultured in YPD medium, EVs will be recovered in the sediment fraction when the culture supernatant is subjected to ultracentrifugation at 100,000 × g [29]. This study confirmed by electron microscopy that *Cn* EVs were recovered in the 100,000 × g sediment fraction (Fig 13B). EG was detected in both the supernatant and sediment fractions of KO strain but was barely detected in both fractions of WT strain (Fig 13C). AEGs were detected in the sediment fraction but not the supernatant fraction of KO strain (Fig 13D), suggesting that AEGs are exclusively exported to the extracellular milieu via EVs. The colony-scraping method was also used to prepare EVs from *Cn* colonies on YPD agar plates [30]. EVs were recovered in the sediment fraction when colony suspension was subjected to ultracentrifugation at 100,000 × g. This method also revealed that EG (Fig 13E) and AEGs (Fig 13F) were recovered in the EV fraction of KO strain but not in that of WT and RE strains. The molecular species of AEGs in EVs were determined to be 16:0-EG, 18:0-EG, 18:1-EG, and 18:2-EG by LC-ESI MS/MS (Fig 13D and 13F). These results suggest that both EG and AEGs in KO strain can be transported extracellularly via EVs, but only EG, not AEGs, is additionally expelled from the cell by a non-EV pathway, possibly by transporters.

**EG transport by ABC transporters in a budding yeast.** EG, but not AEGs, can be transported out of *Cn* cells by non-EV pathways (Fig 13C and 13D), which may involve specific plasma membrane transporters. To explore the possibility, we used a budding yeast system in which each of the ten ABC transporters was deleted in an EG-accumulating strain overexpressing EG synthase (Ugt51) [31] and lacking the EG-degrading enzyme (Egh1) [32]. EG was found in the supernatants of all mutant strains lacking ABC transporters except Yor1, indicating that Yor1 is responsible for EG export in the EG-accumulating budding yeast (S12 Fig). Yor1 is an oligomycin resistance ATP-dependent permease and functions as a pleiotropic drug pump at the plasma membrane to clear toxic substances from the cytosol [33]. Although *Cn* possesses several proteins homologous to Yor1, the specific proteins involved in EG transport in *Cn* remain to be identified.

## AEG-dependent cytokine production, Mincle expression, and KO strain clearance from mice

The increase in MIP-2 production in mouse bone marrow-derived dendric cells (BMDCs) by EVs from KO strain is significantly higher than that from WT and RE strains (Fig 14A). Similarly, IL-8 production from human monocyte-derived dendric cells (MoDCs) by EVs from KO strain is markedly higher than those from WT and RE strains (Fig 14B). MIP-2 production was significantly increased when BMDCs were co-cultured with KO strain compared with WT and RE strains. This increase in MIP-2 production induced by KO strain was significantly suppressed when BMDCs from Mincle KO mice were used (S13 Fig). Treatment of mouse BMDCs with chemically synthesized 18:1-EG enhanced the production of MIP-2 and TNF-α; however, when BMDCs from Mincle KO mice were used, the production of these cytokines was significantly decreased compared with BMDCs from WT mice (Fig 14C and 14D). EG did not enhance these cytokine productions in mouse BMDCs. These results suggest that AEGs, but not EG, enhance cytokine production in mouse BMDCs in a Mincle-dependent manner. Chemically synthesized EG at 1 nmol/well and 18:1-EG at 0.04 -1 nmol/well enhanced IL-8 production in human MoDCs (Fig 14E). The amount of IL-8 produced in human MoDCs by 18:1-EG at 0.2 nmol/well was comparable to that induced by TDB at the same concentration under the conditions used. 18:1-EG, TDB, and LPS

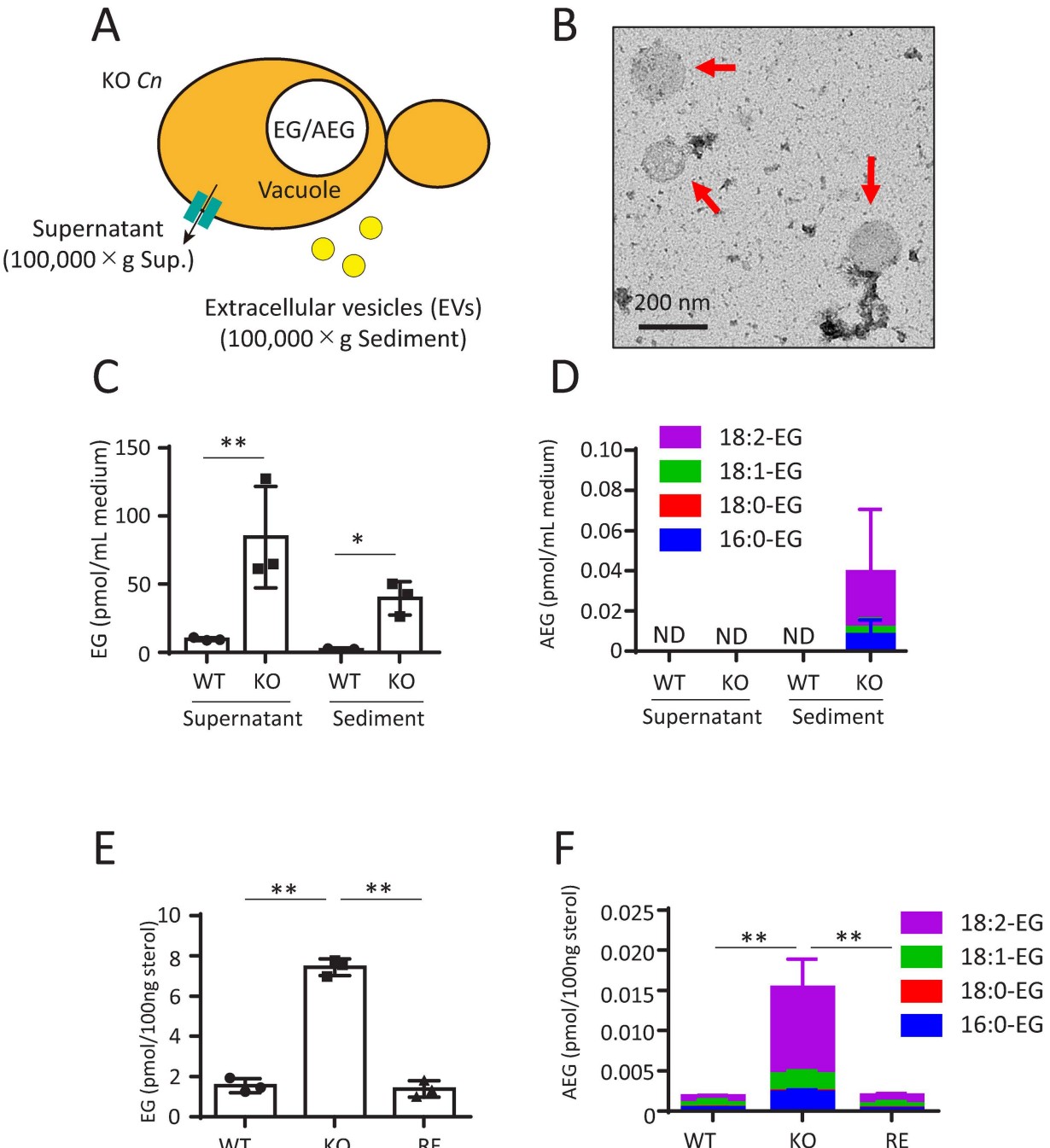

**Fig 13. Extracellular transport of EG and AEGs by EVs.** A, Schematic depiction of EG and AEGs excluded from KO strain. EVs containing EG and AEGs are recovered in the sediment fraction when the culture supernatant of KO strain is subjected to ultracentrifugation at 100,000 x g [29]. B, Transmission electron microscopy (TEM, negative staining) photograph displaying EVs (indicated by red arrows) in the sediment at 100,000 x g. C and D indicate EG and AEG contents in the supernatants and sediments (EV fractions) of WT and KO strain [29]. WT and KO strain were cultured in YPD liquid medium at 30°C for 2 days with shaking. ND, not detected. E and F indicate EG and AEG contents in supernatant and sediments (EV fractions) of WT, KO, and RE strains [30]. WT, KO, and RE strains were cultured in YPD agar medium at 30°C for 1 day. The method for preparing EVs was described in Materials and methods. EG and AEG contents were quantified using LC-ESI MS/MS. Data are presented as Mean±SD (n=3).

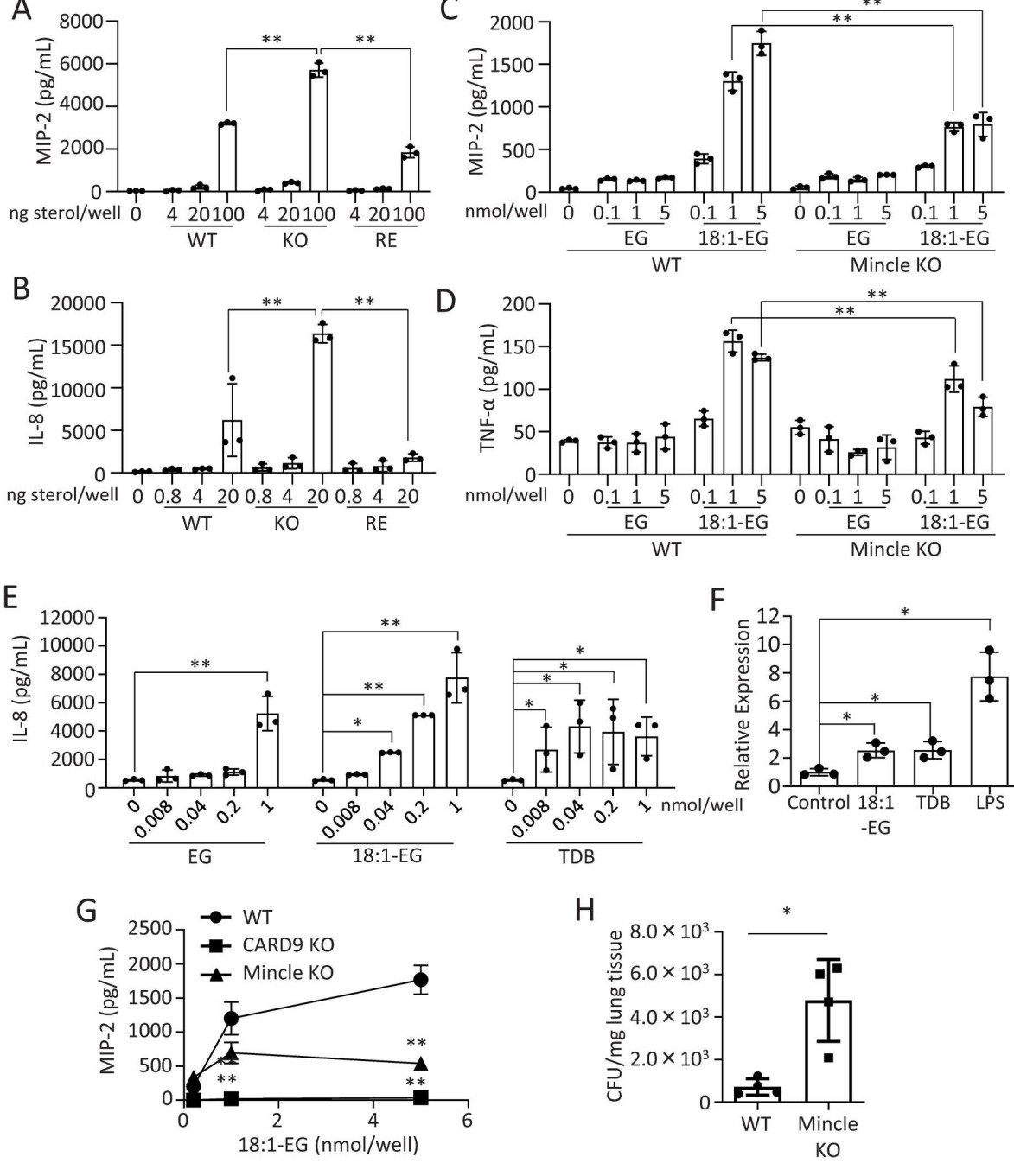

**Fig 14. AEG-dependent cytokine production, Mincle expression, and KO strain clearance from mice.** A, MIP-2 production in mouse BMDCs by EVs from WT, KO, and RE strains. BMDCs from WT mice were exposed to different concentrations (0, 4, 20, 100 ng sterol equivalent EVs per well) of EVs from WT, KO and RE strains for 40 h at 37°C. B, IL-8 production in human MoDCs by EVs from WT, KO and RE strains. Human MoDCs were exposed to different concentrations (0, 0.8, 4, 20 ng sterol equivalent EVs per well) of EVs from WT, KO, and RE strains for 40 h at 37°C. C, MIP-2, D, TNF-α productions in mouse BMDCs by chemically synthesized EG and AEG (18:1-EG). BMDCs from WT and Mincle KO mice were exposed to EG and 18:1-EG at different concentrations (0, 0.1, 1, 5 nmol per well) for 40 h at 37°C. E, IL-8 production in human MoDCs. Human MoDCs were stimulated with EG, 18:1-EG, and TDB (0, 0.008, 0.04, 0.2, 1 nmol per well). F, Mincle expression in mouse BMDCs following stimulation with 18:1-EG (0.1 nmol), TDB (0.1 nmol), and LPS (10 ng/ml) for 40 h at 37°C. Mincle expression was quantified by qPCR. G, MIP-2 production in BMDCs from WT, Mincle KO,

and CARD9 KO mice. BMDCs were stimulated with 18:1-EG at various concentrations (0, 0.1, 1, 5 nmol per well). ELISA was conducted to determine MIP-2 (A, C, G), IL-8 (B, E), and TNF-α (D) productions. Data are presented as Mean ± SD (n = 3). H, CFU in the lung on day 3 after infection of KO strain. WT and Mincle KO mice were intranasally administered $5 \times 10^5$ KO strain cells. CFU was determined by lung *Cn* burden analysis, as shown in Materials and methods. The values represent CFU per mg of lung tissue. Data are presented as Mean ± SD (n = 4).

enhanced the expression of Mincle in mouse BMDCs at the mRNA level (Fig 14F); however, in contrast to TDB, cytokine production in mouse BMDCs by LPS occurs independently of Mincle (S14 Fig). AEG-dependent production of MIP-2 (Fig 14C and 14G) and TNF-α (Fig 14D) was significantly reduced but still detectable in BMDCs from Mincle KO mice. In contrast, MIP-2 production was completely absent in BMDCs from CARD9 KO mice (Fig 14G). CARD9 plays a crucial role in antifungal immunity by linking pathogen recognition receptors, such as CLRs, to downstream signaling cascades that activate the immune system [34]. These results suggest that AEG-dependent inflammatory cytokine production relies entirely on CARD9 and that AEGs trigger inflammatory responses via a multi-receptor system, in which Mincle plays a central role. CLRs other than Mincle, such as Dectin-1 and Dectin-2, did not interact with AEGs under the conditions tested (Fig 11B), indicating that AEGs may mediate cytokine production through unidentified CLRs or non-CLR pathways, in addition to Mincle. Notably, the number of KO strain in lung tissue from Mincle KO mice was significantly higher than WT mice on day 3 after infection (Fig 14H), suggesting that Mincle contributes to the clearance of KO strain at an early stage of infection. In the mouse infection model, KO strain lost its virulence, regardless of the presence or absence of Mincle (S15 Fig).

### Enhanced cytokine production and Mincle expression in mice by ASiG administration

To study the *in vivo* activation of mouse Mincle by ASGs, we used ASiG, which activates Mincle to the same extent as other ASG (Fig 11D). ASiG (100 ng/mouse) was intranasally injected into the lungs, and cytokine production, Mincle expression, and CFU in the lung tissues were analyzed on day 3 after infection with *Cn* according to the scheme (Fig 15A). The results showed significant increases in the production of TNF-α (Fig 15B), MIP-2 (Fig 15C), and IL-17A (Fig 15D) in lung tissue after ASiG administration compared with the control groups that did not receive ASiG. The increase in cytokine production by ASiG administration was attenuated in Mincle KO mice (Fig 15B and 15C). Like the stimulation of mouse DCs with AEGs (Fig 14F), Mincle expression at the mRNA level increased in lung tissue after ASiG administration (Fig 15E). CFU in lung tissue was significantly decreased by ASiG administration (Fig 15F). These results indicate that ASiG administration enhances cytokine production in mouse lung tissue and that this effect partially depends on Mincle.

## Discussion

### Dysfunction of autophagy in KO strain

It has been reported that KO strain, compared with WT or RE strains, exhibits lower survival rates under conditions of nutrient starvation and oxygen deficiency *in vitro* [10] and significantly lower virulence in mice [8]. The reasons for this are not well understood. Our current research suggests that autophagy dysfunction in KO strain at 37°C is a molecular-level explanation for the reduced survival and virulence of KO strain. Several studies have already reported that autophagy plays an essential role in the survival and virulence of *Cn*. For example, the pathogenicity of *Cn* in mice is markedly weakened by disrupting any of the *ATG1*, *ATG7*, *ATG8*, or *ATG9* genes [15]. The disruption of *ATG* genes in *Cn* reduces its pathogenicity in a *Galleria mellonella* infection model [35]. We found that the deletion of EGCrP2/Sgl1 reduced the expression of several *ATG* genes in *Cn* at 37°C (Fig 3). Notably, there was a significant decrease in the expression of *ATG* genes that encode key components of the Atg8/Atg12 ubiquitin-like conjugation systems, potentially indicating a failure of these systems. This reduction included genes encoding Atg8 (which plays a crucial role in the formation of autophagosomes), Atg4 (an enzyme that cleaves the carboxy-terminal arginine residue of Atg8), Atg3 (an E2 enzyme involved in the activation of Atg8), and Atg5/Atg12 (which, after conjugation with Atg16, functions similarly to an E3 enzyme in transferring

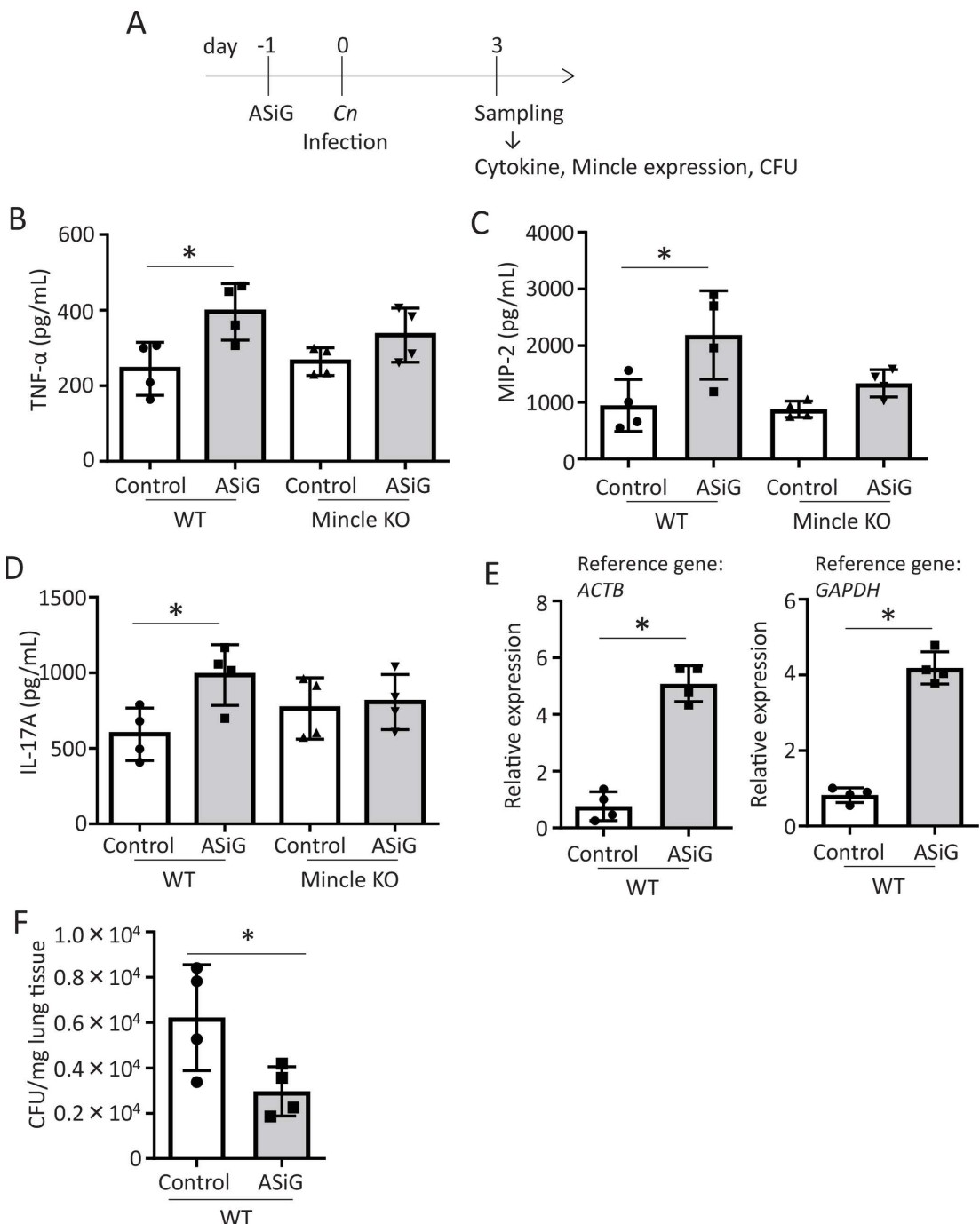

**Fig 15. Cytokine production, Mincle activation, and CFU in mouse lung tissues after administration of ASiGs.** A, Experimental design. According to the experimental design, WT and Mincle KO mice (4 mice per group) were intranasally administered 100 ng of ASiGs in 50 μL of PBS containing 1% DMSO. PBS containing 1% DMSO without ASiGs was used for the control. B, TNF-α, C, MIP-2, and D, IL-17A productions in mouse lung homogenates. ELISA quantified cytokine concentrations in homogenates. E, Mincle expression in lung tissues. Mincle expression was assessed through qPCR using *Actb* (right) and *Gapdh* (left) as reference genes. F, CFU in lung tissues. CFU was determined by lung *Cn* burden analysis, as shown in Materials and methods. Data are presented as Mean ± SD (n = 4). The experiment was performed twice and yielded consistent results, with one dose of ASiG administered per experiment.

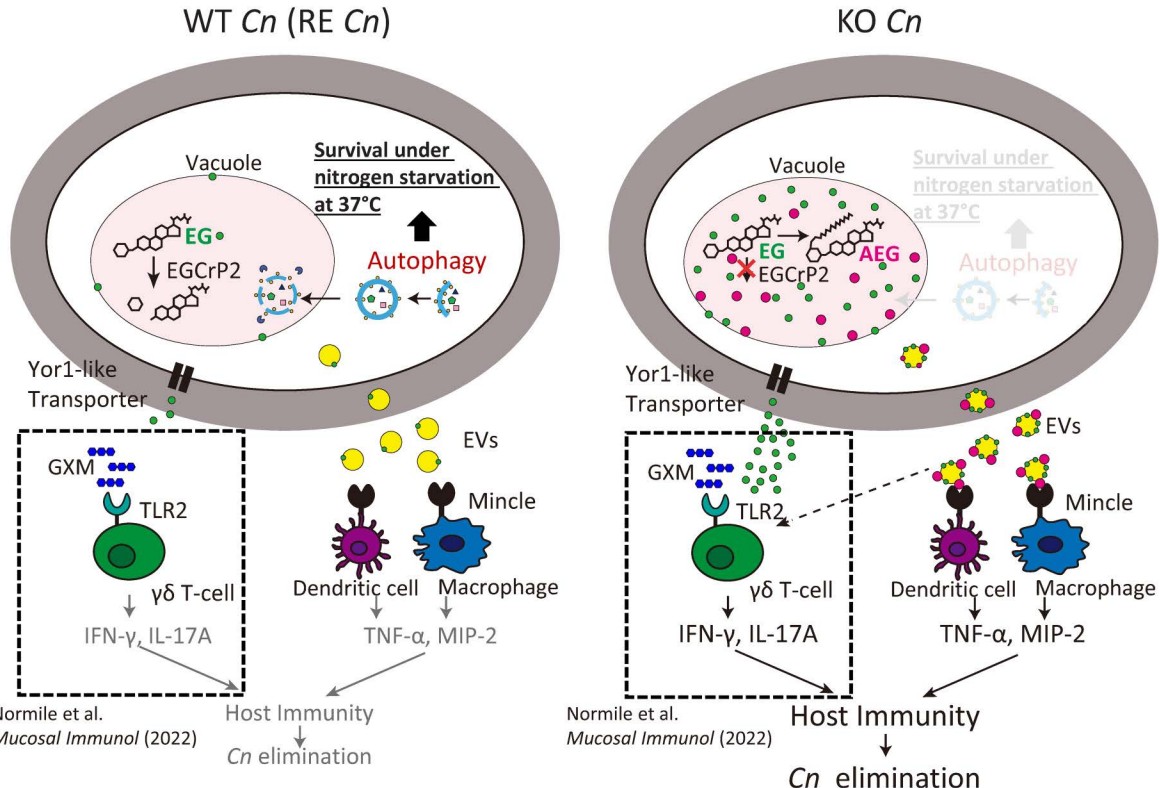

**Fig 16. Visualization of the impact of EGCrP2/Sgl1 deficiency on *Cn* and mouse immunity.** In *Cn*, EG is usually degraded in vacuoles by EGCrP2/Sgl1. However, EG and its acylated derivatives, AEGs, accumulate in KO strain. This accumulation disrupts autophagy, leading to cell death of KO strain under nitrogen starvation conditions. AEGs are trafficked via EVs, whereas EG is transported through EVs and non-EVs pathways. EVs derived from KO strain enhance cytokine production through Mincle activation. EG promotes GXM-mediated TLR2 signaling in γδ T cells, producing cytokine and activating immune cells [42]. Ultimately, the activated host immune system eliminates KO strain from the mouse. The dotted squares in the figure are not the results of this study but a summary of the report by Normile et al [42]. Yellow circles, EVs; green circles, EG; red circles, AEGs; dark blue hexagons, glucuronoxylomannan (GXM).

Atg8 from Atg3 to phosphatidylethanolamine, PE) [13]. Consequently, the decreased expression of these genes led to a significant delay in the entry of GFP-Atg8 into the vacuole in KO strain at 37°C (Figs 4 and 5), leading to cell death of KO strain under nitrogen starvation conditions (Figs 2B-D and 7).

## Activation of Mincle by EG and AEGs

Trehalose 6,6'-dimycolate (TDM), a glycolipid found in the cell wall of *Mycobacterium tuberculosis*, exhibits immunostimulatory activity. Mincle was reported to recognize TDM directly [36]. Further studies have shown that the structure of TDM itself is not essential for binding to Mincle, i.e., substituting mycolic acid for fatty acid and trehalose for glucose in TDM had little effect on binding activity to Mincle [20]. Structural analysis of Mincle revealed that this CLR has two sugar-binding and two lipid-binding pockets, and glycolipids that interact with three pockets function as potent ligands for Mincle [19,20,28]. A binding model for mouse Mincle either with EG or AEG suggests that the glucose of both glycolipids can interact with the EPN motif (sugar-binding pocket) via the site 1 $Ca^{2+}$ ion-mediated binding network (Fig 12). The sterol moiety of both glycolipids can interact with a lipid-binding pocket (S11 Fig). Compared with EG, AEGs have a fatty acid linked to glucose. This fatty acid can interact with another lipid-binding pocket via the site 3 $Ca^{2+}$ ion-mediated binding network (Fig 12). This model supports the results from ligand-Mincle activating experiments (Fig

11), showing that AEGs (ASGs) are a more potent ligand for mouse Mincle than EG (SGs). However, crystallographic analysis of the Mincle/AEG and Mincle/EG complexes is required to elucidate the details of the binding mode of Mincle and its novel ligands.

### Role of Mincle in the *Cn* mouse infection model

Mincle deficiency significantly reduced 18:1-EG-induced production of MIP-2 and TNF-α in mouse BMDCs and increased CFUs in mouse lungs during early infection (Fig 14C, 14D and 14H), indicating that the 18:1-EG-mediated immune response depends on Mincle. Furthermore, the ASG-Mincle immune activation functions in terms of increased cytokine production and *Cn* clearance when WT strain is administered with ASiG. This result indicates that Mincle activation by ASGs can enhance immune responses without autophagy defects. However, in the current mouse infection model, KO strain lost its virulence, regardless of the presence or absence of Mincle (S15 Fig). The metabolic dysfunction of EG and AEG led to autophagy dysfunction in KO strain under nitrogen starvation at 37°C (Figs 3-6), ultimately leading to cell death (Figs 2 and 7). In the mouse infection model, KO strain CFU in the lungs of WT mice on day 3 is less than 5% compared with WT and RE strains CFU (Fig 1B). These results indicate that autophagy defect is the primary cause of why KO strain lost its virulence, even in the Mincle KO mice.

In a mouse model of *Candida albicans* infection, Mincle deficiency did not affect survival time [37]. However, it was significantly shortened when multiple CLRs, such as Mincle/Dectin-1, Mincle/Dectin-2, or Mincle/Dectin-1/ Dectin-2, were knocked out, indicating that other CLRs can compensate for the loss of Mincle. Similarly, the AEG-dependent immune responses observed in our study suggest the involvement of unidentified CLRs or non-CLR pathways in addition to Mincle (Fig 14C, 14D and 14G). This compensatory mechanism may partly explain why Mincle-deficient mice showed no significant difference in survival time compared with wild-type mice in the KO strain infection model.

### Role of EVs in excluding EG and AEGs accumulated in KO strain

In *Cn*, EVs were first identified as carriers of polysaccharides across the fungal cell wall [38]. Subsequent studies showed that fungal EVs transport diverse substances, including proteins, lipids, nucleic acids, toxins, and other molecules [39]. Despite their significance, the mechanisms underlying the biosynthesis and transport of fungal EVs still need to be better understood. One of the most intriguing findings in this study is that EVs can transport EG and AEGs, which accumulate in the vacuoles of KO strain, out of the fungal cells (Fig 13). This mechanism resembles mammalian cellular responses where potentially toxic molecules are packaged into vesicles and expelled into the extracellular space to prevent harmful accumulation, as seen in lysosomal storage disorders [40]. Similarly, when lysosomal function is impaired, mitochondria destined for lysosomal degradation are expelled in large EVs through the endosomal pathway [41]. These systems are considered to be adaptive responses to cellular stress and impaired degradation mechanisms, ensuring the cell can still function optimally under unfavorable conditions.

### Host immune response by EG and AEGs (ASGs)

Normile *et al.* reported that the administration of KO strain protected mice from subsequent *Cn* infection even in the absence of CD4[+] T cells [42]. This vaccine effect may be due to the increased production of IFN-γ and IL-17A by γδ T cells since the neutralization of these cytokines or the absence of γδ T cells results in a loss of protection against WT strain. TLR2[-/-] mice did not produce IL-17A in response to KO strain administration. They were no longer protected from the WT strain challenge, suggesting that the EG accumulated in KO strain may stimulate γδ T cells to produce IFN-γ and IL-17A via TLR2. Their study also suggested a role for EG as an adjuvant for glucuronoxylomannan (GXM) in stimulating TLR2 signaling; however, the direct target of EG in adjuvant effects remains unsolved.

No studies except our study examined the role of glycolipids accumulated in KO strain in the innate immune system, particularly regarding the involvement of CLRs, which may interact with fungal glycolipids [43]. A key finding of our study is that while Mincle plays a limited role in WT strain infection in a mouse model [44], its involvement is significantly enhanced in KO strain due to the accumulation of AEGs, contributing more prominently to the clearance of KO strain from the mouse at day 3 (Fig 14H).

Administration of ASiG enhanced cytokine production and Mincle expression in mice (Fig 15). ASG is widely distributed in plants, including eatable grains [45], which highlights the importance of investigating the effects of dietary ASGs, including ASiG, on the human innate immune system via Mincle.

### Physiological relevance of EG, AEGs, and ASGs in vivo

The amount of AEGs in KO strain was much lower than that of EG when cultured at 25°C or 30°C. However, at 37°C, the amount of AEGs increased significantly, reaching approximately 1 pmol/mg DCW (Fig 10). This temperature-dependent increase of AEGs in KO strain suggests that higher AEG levels may be relevant during infection.

We compared MIP-2 production in BMDCs from WT and Mincle KO mice when co-cultured with WT, KO, and RE strains. The results showed that MIP-2 production was significantly higher in response to KO strain than WT and RE strains. However, this increase in MIP-2 production induced by KO strain was significantly suppressed when BMDCs from Mincle KO mice were used (S13 Fig). These findings suggest that the amount of AEG produced by co-cultured KO strain is sufficient to activate BMDCs to produce cytokines in a Mincle-dependent manner under experimental conditions.

To evaluate the physiological relevance of ASGs in vivo, we administered ASiG (an AEG analog) nasally to mice. ASiG activated Mincle to the same extent as other ASGs (Fig 11D). Even a minimal dose of ASiG (100 ng/mouse, 114 pmol as 18:3-SiG/mouse) resulted in a significant increase in Mincle expression, cytokine production, and decreased CFU in lung tissue compared to the untreated control group (Fig 15). These findings indicate that ASiG is a potent Mincle-related immunostimulatory glycolipid in vivo. Since the sterol moieties of ASGs did not influence Mincle stimulation activity (Fig 11D, 11G and 11H), Cn-derived AEGs may similarly have physiological relevance in vivo.

Mouse Mincle responds to ASGs (AEGs) but shows little reactivity to SGs (EG). In contrast, the human Mincle reacts with both EG and AEGs (Fig 11F-H), further supporting the relevance of these Cn-derived glycolipids in human immune activation and their potential as targets for antifungal drugs.

### Why does KO strain lose virulence in a mouse infection model?

Fig 16 illustrates the working hypothesis explaining the significantly reduced pathogenicity of KO strain in a mouse infection model compared with WT and RE strains (Fig 1). In WT strain, EG is degraded by the vacuolar-localized enzyme EGCrP2/Sgl1 (Fig 8). In contrast, EG accumulates in the vacuole of KO strain, where some are converted to AEGs (Fig 9). In KO strain, the downregulation of ATG genes leads to dysfunctional autophagy (Figs 3–6). This autophagy defect is the leading cause of increased lethality in KO strain at 37°C under nitrogen starvation conditions (Fig 7). Meanwhile, the accumulated AEGs in the vacuole are transported out of the fungus by EVs (Fig 13), where they act as a ligand for Mincle, promoting the production of cytokines such as TNF-α and MIP-2, which contributes to the clearance of KO strain at an early stage of infection (Fig 14). EG, but not AEGs, can be transported extracellularly by Yor1-like ABC transporters and can function as an adjuvant for GXM-mediated TLR2 activation in γδ T cells to enhance the production of IFN-γ and IL-17A [42]. In this context, the potential stimulation of γδ T cells by ASiGs deserves further investigation since ASiGs administration in mice increased IL-17A production (Fig 15D). Notably, acylated cholesterol α-glucoside (αCAG) from *Helicobacter pylori* is a ligand for Mincle and is crucial for T cell priming [46].

## Conclusion

The metabolism of EG in the vacuoles by EGCrP2/Sgl1 is crucial for the survival and virulence of *Cn* at 37°C. In KO strain, the accumulation of EG and AEGs leads to dysfunctional autophagy under nitrogen starvation conditions at 37°C, leading to cell death. To avoid cellular toxicity, *Cn* exports these glycolipids via EVs. However, once released, AEGs activate Mincle in mice, while in humans, both EG and AEGs activate Mincle. AEG-dependent Mincle activation enhances cytokine production, contributing to the clearance of KO strain from mice. The findings of this study, along with previous research [10,42], are expected to aid in the development of novel antibiotics and vaccines aimed at combating IFDs caused by *Cn* and *Aspergillus fumigatus* [26].

## Materials and methods

### Ethics statement

All animal procedures received approval from the Recombinant DNA Experiments Safety Committee and Animal Care and Use Committee of Kawasaki Medical School (approval numbers: 21–26, 21–071, 21–080, 23-035). These procedures were carried out strictly according to the ARRIVE (Animal Research: Reporting of In Vivo Experiments) guidelines.

### Materials

Cholesteryl glucoside (#28609) was obtained from Sigma-Aldrich. Stigmasterol β-glucoside (#700160), 18:1-stigmasterol β-glucose (#700167), and trehalose 6,6'-behenate (#890808) were sourced from Avanti Polar Lipids. Sitosterol β--glucoside (#1117) and acylated sitosterol β-glucoside (ASiGs, #1118) were acquired from MATREYA LLC. pGWKS2 and pGWKS7 were provided by James Fraser (Addgene plasmid # 139410; http://n2t.net/addgene:139410; RRID: Addgene_139410 and Addgene plasmid # 139414; http://n2t.net/addgene:139414; RRID:Addgene_139414) [47].

### Fungal strains and culture conditions

*Cryptococcus neoformans* var. *grubii* serotype A strain H99 (ATCC 208821) was purchased from the American Type Culture Collection (ATCC). A *SGL1*-disrupted *C. neoformans* mutant strain (*sgl1Δ*, KO) was generated as described previously [7]. To create a revertant strain (*sgl1Δ::SGL1*, RE), a *SGL1*-expressing construct was reintroduced into the KO strain. *Cn* was cultured in YPD medium (2% glucose, 2% peptone, 1% yeast extract) with continuous shaking (150 rpm) at a specified temperature.

### Mice

Mincle-deficient mice (Mincle KO mice) underwent backcrossing for at least nine generations with C57BL/6 mice [17]. Professor H. Hara of Kagoshima University generously provided CARD9-deficient mice (CARD9 KO mice)[48]. C57BL/6J mice (WT) were obtained from CLEA, Japan. The mice were housed under standard conditions, maintaining a controlled environment of 22±2°C with a 12-h light/12-h dark cycle (lights on at 8 A.M.). Hosting consisted of 5–6 mice per cage, with unrestricted access to both food and water.

### Cells

The 2B4-nuclear factor of activated T cells (NFAT)-GFP reporter cells expressing various C-type lectin receptors were prepared as described previously [49]. Secreted alkaline phosphatase (SEAP) reporter HEK293 cells expressing the mouse Mincle were purchased from InvivoGen.

### Mouse infection experiments with *Cn*

*Cn* infection experiments were conducted using previously described methods [8] with minor modifications. Female C57BL/6J mice aged five weeks were used for this study. Mice were anesthetized with an intraperitoneal injection of a

mixture of anesthetic agents (medetomidine [0.75 mg/kg], butorphanol [5.0 mg/kg], and midazolam [4.0 mg/kg]). WT and Mincle KO mice (6 per group) were infected intranasally with $5 \times 10^5$ cells/50 μL of WT, KO, or RE strain in sterilized PBS. Mice were monitored twice daily, and those showing signs of distress or becoming moribund were humanely sacrificed using carbon dioxide.

## Histological analysis

Lung tissues were fixed with 4% paraformaldehyde in PBS. The paraffin-embedded sections were stained with hematoxylin and eosin staining. Comprehensive lung section images were captured using a BZ-X800 microscope (Keyence).

## Lung *Cn* burden analysis

Lung homogenates, suitably diluted with PBS, were plated on YPD agar plate supplemented with 100 μg/mL of ampicillin and incubated at 30°C for 1–2 days. Subsequent colony counting facilitated the determination of colony-forming units (CFUs).

## Growth and viability of *Cn*

*Cn* was grown in YPD medium for specific durations: 48 h (day 2), 72 h (day 3), 96 h (day 4), and 168 h (day 7), at 25, 30, and 37°C with shaking at 150 rpm. The optical density at 600 nm (OD600) was measured using the GENESYS 10S UV-Vis spectrophotometer (Thermo Fisher Scientific). Cell viability was examined by staining with phloxine B [21]. Briefly, $5.0 \times 10^6$ cells were washed with PBS twice and resuspended in PBS containing 500 μg/mL phloxine B (Fujifilm Wako Pure Chemical Industries). After 5 seconds of sonication, the cells underwent a single wash with PBS before resuspending in 1 mL of PBS. Cells stained with phloxine B were visualized utilizing a DMi8 fluorescence microscope (Leica microsystems) equipped with a ×100, 1.4 numerical aperture (NA) Plan Apochromat oil objective lens (Leica). The percentage of non-viable cells was quantified through analysis using Attune NxT Flow Cytometer (Thermo Fisher Scientific).

## RNA sequencing (RNA-seq)

Total RNA was isolated from *Cn* after 72 h (day 3) of incubation at 30 and 37°C (n = 1 for each WT, KO, RE strain at both temperatures). RNA isolation was performed using the SV total RNA Isolation System (Promega) and ReliaPrep RNA Miniprep System (Promega). The quality of RNAs from WT, KO, and RE strains cultured at 30°C and 37°C was checked using RNA Integrity Numbers (RIN) values as an indicator. The results showed that the RIN values of all RNAs were above 7.0, indicating that they had sufficient purity to be used for RNA-seq analysis (S16 Fig). RNA-seq was conducted by GENEWIZ, with data analysis employing *HISTAT*2, Samtools, String-Tie, and Ballgown [50]. The *Cn* genome index file and general feature format file were obtained from Fungi DB [51]. K means clustering analysis was performed utilizing iDEP.92 [52], and gene ontology (GO) annotation and enrichment analysis were conducted using DAVID [53]. cDNA synthesis was done using the PrimeScript RT Reagent Kit with gDNA Eraser (Perfect Real Time) (Takara-Bio).

## Quantitative PCR (qPCR)

*Cn* total RNA was isolated using the SV total RNA Isolation System (Promega) and ReliaPrep RNA Miniprep System (Promega). Mouse total RNA was isolated using the RNeasy Plus Universal Mini kit (QIAGEN). RIN values of RNAs prepared were above 7.0. cDNA synthesis was done using the PrimeScript RT Reagent Kit with gDNA Eraser (Perfect Real Time) (Takara-Bio). qPCR was performed using Mx3000p qPCR System (Agilent Technologies) for *Cn* and QuantStudio1 qPCR System (Thermo Fisher Scientific) for mouse with TB Green Premix Ex Taq II (Tli RNaseH Plus) (Takara-Bio). Relative mRNA expression levels were determined using the PCR efficiency-corrected method [54], normalized to actin and GAPDH mRNA levels. Oligonucleotide primer sets are shown in S2 Table.

## Autophagy evaluated by translocation of GFP-Atg8 into vacuoles

This experiment was performed according to the method described in [24]. ORF of *Cn ATG8* was inserted into pGWKS2, a GFP-expressing vector for *Cn*, to generate a GFP-Atg8-expressing construct. The nourseothricin-resistance gene (*NAT*) of pGWKS2 was replaced with neomycin resistance gene (*NeoR*) by inverse PCR using primers described in S2 Table. After digesting with Bae I (New England Biolab), a GFP-Atg8-expressing construct was introduced into WT, KO, and RE strains by biolistic transformation, as described in [55]. GFP-Atg8-expressing strains were selected by YPD agar plates containing 100 µg/mL of G418. GFP-Atg8 WT, KO, and RE strains were cultured in 20 mL YPD medium in 50 mL flasks at 37°C with 150-rpm shaking for 4 days. The localization of GFP was examined daily using a DMi8 fluorescence microscope using ×100, 1.4 NA Plan Apochromat oil objective lens. Fluorescent signals of GFP were observed using an FITC filter set with a 460–500-nm excitation filter, dichromatic mirror at 505 nm, and 512–542 nm emission filter. Among GFP-positive cells, the percentage of cells with GFP localized in the vacuole was calculated by counting at least 100 cells per each measuring time. To induce autophagy as described previously [14], GFP-Atg8 WT, KO, and RE strains cultures were started as $OD_{600} = 0.1$ and placed in 3 mL YPD medium at 30°C with 150-rpm shaking for 15 h. Cells were washed three times with PBS and then transferred to 3 mL of nitrogen starvation medium, SD-N [0.17% Difco Yeast Nitrogen Base w/o amino acids and ammonium sulfate (Becton, Dickinson, and Company), 2% glucose]. Cells were cultured at 37°C and localization of GFP-Atg8 was observed after 0, 1-, 2-, 3-, and 4-hour induction.

## Autophagy evaluated by cleaving GFP-Atg8

Autophagy progression was also confirmed by cleaving GFP-Atg8 through immunoblot analysis using an anti-GFP antibody [GFP (D5.1) Rabbit mAb #2956, Cell Signaling Technology, RRID: AB_1196615]. GFP and GFP-Atg8 were detected by Luminata Forte Western HRP substrate (Merck Millipore) after incubation with anti-rabbit IgG, HRP-linked antibody (#7074, Cell Signaling Technology, RRID: AB_1099233). The selection and culture of GFP-Atg8-expressing *Cn* cells were done using the methods described above. *Cn* samples (0.5 mL) in SD-N medium were collected after 0, 1-, 2-, 3-, and 4 hours and then stored at -80°C. Each sample was lyophilized and crushed with BioMasher II (Nippi). Then, it was suspended in 200 µL of RIPA Lysis and Extraction Buffer (Thermo Fisher Scientific) containing a protease inhibitor cocktail (cOmplete, Roche) and 1 mM PMSF (Nacalai Tesque). After sonication for 10 s, cell lysate was centrifuged at 17,000 × g for 10 min at 4°C. The protein concentration in the supernatant was measured as described below, and equal amounts of cellular protein were separated by SDS-PAGE using a Mini-PROTEAN TGX Gel (BioRad) and subsequently blotted onto a PVDF membrane by Trans-Blot Turbo (BioRad). The PVDF membrane was stained by Ponceau-S (Beacle, Inc).

## Detection of dead *Cn* cells under autophagy-inducing conditions

To investigate the relationship between autophagy deficiency and cell death, WT, KO, and RE strains were cultured at 37°C under both autophagy-inducing (SD-N medium) and nutrient-rich (YPD medium) conditions, followed by phloxine B staining to detect dead cells. Each strain was cultured in 3 mL of YPD at 30°C for 2 days. The cultures were subcultured into fresh 3 mL YPD and incubated at 30°C for another 4 days. The cultures were then adjusted to an OD600 of 0.1 using fresh YPD and incubated at 30°C for 16 hours. Cells were washed three times with PBS, transferred to 3 mL of SD-N medium, and incubated at 37°C. Samples were collected for phloxine B staining at 0, 2, 4, and 8 hours. Parallel cultures in YPD at 37°C served as controls. phloxine B-stained cells were observed under a fluorescence microscope with a ×40, 0.7 NA PL Fluotar objective lens (Leica). More than 150 cells were analyzed for each time point, and the percentage of phloxine B-positive cells was quantified.

## Vacuole staining of *Cn* with FM4–64

Vacuoles of GFP-Atg8 expressing WT, KO, and RE strains were co-stained with 16 µM of FM4–64 and observed using a Texas Red filter set.

 PLOS Pathogens

### Intracellular localization of EGCrP2/Sgl1 in *Cn*

*Cn* EGCrP2/Sgl1 ORF was inserted into the mRuby-expressing vector (pGWKS7) to generate the mRuby-EGCrP2/Sgl1 construct. The nourseothricin-resistance gene (NAT) of pGWKS7 was replaced with the neomycin resistance gene (NeoR) by inverse PCR using primers described in S2 Table. After digesting with Bae I (New England Biolab), the mRuby- or mRuby-EGCrP2/Sgl1 expressing construct was introduced into *Cn* by biolistic transformation using the previously described method [55]. Transfectants were selected by YPD agar plates containing 100 µg/mL of G418 (Nacalai Tesque). Fluorescence microscopy using a DMi8 fluorescence microscope with a × 100, 1.4 NA Plan Apochromat oil objective lens facilitated observation of mRuby. Vacuoles were visualized by the weak base, quinacrine, trapped in acidic cellular compartments such as fungal vacuoles [56]. The fluorescent signals of mRuby were observed using a Texas Red filter set comprising a 540–580-nm excitation filter, dichromatic mirror at 585 nm, and 592–668-nm emission filter. The fluorescence of quinacrine was observed using a FITC filter set composed of a 460–500 nm excitation filter, dichromatic mirror at 505 nm, and 512–542-nm emission filter. *Cn* expressing mRuby alone was used as a control.

### Extraction and TLC analysis of glycolipids

Total lipids were extracted from WT, KO, and RE dry cells using chloroform/methanol (2/1, v/v). The glycolipids were then separated by TLC using chloroform/methanol/water (65/16/2, v/v/v) as a developing solvent and visualized using orcinol sulfate.

### LC-ESI MS/MS analysis of EG and AEGs

For quantifying EG and AEG, total lipids were extracted from 4 mg of *Cn* dry cells by sonication for 10 min with 300 µL of chloroform/methanol (2/1, v/v), containing 10 µM cholesterol glucoside and acylated sitosterol β-glucoside as internal standards. After centrifugation (13,500 × g for 5 min) to remove cell debris, the supernatant was augmented with 75 µL of distilled water. The organic phase was applied to LC-ESI MS/MS. EG and AEG were separated through reverse-phase chromatography using an InertSustain C18 column (2.1 × 150 mm, 5 µm, GL Sciences) with a binary solvent gradient at a 200 µL/min flow rate. EG and AEG quantifications were achieved through multiple reaction monitoring (MRM), with specific parameters detailed in S3 Table. The chromatographic peaks obtained were analyzed with Multi Quant software 3.0.1 (SCIEX).

### TLC and GC analyses of AEGs

Total lipids were extracted from KO strain using chloroform/methanol (2/1, v/v). AEGs were separated using a Sep-Pak plus silica cartridge (Waters). Briefly, the total lipids were dissolved in chloroform and applied to the column. The elution process involved chloroform, chloroform/acetone (8/2, v/v), acetone, and methanol. The resulting lipids were reconstituted in chloroform/methanol (2/1, v/v) and applied to a Silica Gel 60 TLC, which was developed with chloroform/methanol/water (65/16/2, v/v/v), and visualized through orcinol sulfate staining. The fraction containing AEGs, extracted using chloroform/acetone (8/2, v/v), underwent GC analysis following the release of fatty acid moieties through metanalysis, as described previously [57]. The characteristic fragment ions of AEGs were detected by enhanced product ion mode of LC-ESI MS/MS setting with collision energy (CE) at 30 or 70.

### Protein assay

The protein content was determined using a 660-nm Protein Assay reagent (Thermo Fisher Scientific) with bovine serum albumin (Thermo Fisher Scientific) as a standard.

### Chemical synthesis of EG and AEGs (16:0-EG and 18:1-EG)

EG and two 6-O-acylated EGs (AEGs) were synthesized starting from D-glucose over five and six steps, respectively. First, D-glucose was benzoylated to give D-glucose pentabenzoate, which was treated with hydrazine acetate in DMF to afford

the lactol derivative. Subsequently, the reaction of the lactol with trichloroacetonitrile and 1,8-diazabicyclo[5.4.0]-7-undecene in dichloromethane gave the glucosyl trichloroacetimidate donor at a good yield. The glucose donor was then glycosidated with ergosterol in the presence of catalytic amounts of trimethylsilyl trifluoromethanesulfonate in dichloromethane at -20 °C, giving EG at a 77% yield. Subsequent removal of benzoyl groups under Zemplén conditions generated EG at a quantitative yield. EG was then subjected to regioselective acylation at the C6 position of the glucose residue. The treatment of EG with palmitoyl chloride or oleoyl chloride in the presence of N,N-diisopropylethylamine, and dimethyl tin dichloride in THF delivered ergosterol 6-O-palmitoyl-β-D-glucopyranoside (yield: 43%) or ergosterol 6-O-oleoyl-β-D-glucopyranoside (yield: 29%), respectively. The chemically synthesized EG and AEG structures are shown in S10 Fig.

### Preparation of mouse bone marrow-derived dendric cells (BMDCs)

BMDCs were prepared by the method described previously [58]. WT and Mincle KO mice were euthanized using carbon dioxide, and femurs and tibiae were separated and disinfected with 70% ethanol for 2 min. Bones were transferred to PRMI-1640 in perti dish, and bone ends were cut with sterile scissors. Bone marrows were flushed using a 27G 3/4 needle syringe and collected by pipetting to 50 mL tube. To remove red blood cells, cells were treated with ACK (Ammonium-Chloride-Potassium) lysing buffer and filtrated through a 70 μm cell strainer (CORNING). BMDCs were obtained from $1 \times 10^7$ cells of bone marrows after culture in 10 mL of RPMI-1640 supplemented with 10% FBS, 20 ng/mL murine GM-CSF (PeproTech) at 37°C with 5% $CO_2$ for 7 days. At day 3, additional 5 mL RPMI-1640 supplemented with 10% FBS, 40 ng/mL murine GM-CSF.

### Preparation of human monocyte-derived dendritic cells (MoDCs)

MoDCs were prepared by the method described previously [59]. Peripheral blood mononuclear cells were isolated from the peripheral blood of healthy donors by Lymphocyte Separation Solution (d = 1.077) (Nacalai Tesque) density gradient centrifugation. CD14+ monocytes were purified from peripheral blood mononuclear cells using anti-human CD14 MicroBeads (Miltenyi Biotec). MoDCs were obtained from CD14+ monocytes after culture in RPMI-1640 supplemented with 10% FBS, 10 ng/mL human GM-CSF (PeproTech), and 10 ng/mL human IL-4 (PeproTech) for 7 days.

### *In vitro* stimulation assay of Mincle

Various SGs and ASGs were dissolved in chloroform/methanol/water (2/1/0.1, v/v/v and 2/1/0, v/v/v, respectively), diluted with 2-propanol, and used to coat wells. The 2B4-NFAT-GFP reporter cells were stimulated at 37°C for 18 h in 5%$CO_2$, and NFAT-GFP activation was quantified by Attune NxT Flow Cytometer and FACS Canto II flow cytometry (BD Biosciences). BMDCs and MoDCs ($1 \times 10^5$ cells per well) were stimulated at 37°C for 2 days in 5%$CO_2$ and the culture supernatants were collected for cytokine quantification through ELISA (R&D SYSTEMS and eBioscience).

### Generation of Mincle docking model with either EG or 18:1-EG

We utilized the Alphafold2 model of mouse Mincle in its apo state (lacking $Ca^{2+}$ and ligand binding). We superimposed it on the bovine Mincle structure (PDB ID: 5KTH) after truncating the first 62 N-terminal residues. Guided by the three $Ca^{2+}$-binding sites in the bovine Mincle [28], a $Ca^{2+}$-bound conformation for the mouse Mincle was derived. Utilizing MOE software (version 2023, Chemical Computing Group, Inc.), we constructed docking models of mouse Mincle with EG and AEG ligands, drawing insights from the ligand-binding model reported for human Mincle [20]. Mouse Mincle's structure, except for the hydrogen atoms, was fixed during this docking procedure.

### Isolation of *Cn* EVs and quantification of their sterols

EVs from liquid culture were isolated from the supernatant of *Cn* culture on YPD-liquid medium [29]. *Cn* (WT, KO, and RE) was cultured in 1 L of YPD medium at 30°C for 2 days. *Cn* cells were removed by centrifugation (4000 × g for 5 min),

and the supernatant was subjected to centrifugation at 15,000 × g for 10 min to remove cell debris and filtrated with a 0.8-μm membrane filter (Merck). The filtrate was concentrated with pressure-based sample concentration (MWCO 100 kDa, Merck) to about 20 mL and subjected to ultracentrifugation (100,000 × g for 60 min) to collect EVs. The precipitate was suspended in 100 μL of sterilized PBS. EVs from agar plate culture were prepared using the methods described previously [30]. *Cn* cells with an approximate OD600 of 0.3 were spread onto a 13-mL YPD agar medium in a petri dish of 10 cm in diameter and cultured at 30°C for 1 day. *Cn* cells were collected from the agar plates by a scraper, suspended in 10 mL of sterilized PBS, and subjected to centrifugation (4,000 × g for 5 min). The supernatant was centrifuged at 15,000 × g for 10 min to eliminate cell debris, followed by ultracentrifugation at 100,000 × g for 60 min to collect EVs. The resulting precipitate, containing EVs, was resuspended in 200 μL of PBS. Sterols within EVs were quantified using the Amplex Red Cholesterol Assay Kit (Thermo Fisher Scientific).

## TEM observation

EVs were visualized using a transmission electron microscope (TEM) with negative staining. Briefly, PBS-containing EVs were placed on grids for 10 min, washed twice with PBS, and then stained with 2% uranyl acetate. EVs were observed using Tecnai 20 TEM (FEI).

## *In vivo* stimulation of Mincle with ASiGs

In the *in vivo* Mincle activation experiment, WT and Mincle KO mice (4 mice per group) were intranasally administered 100 ng of ASiGs in 50 μL of PBS containing 1% DMSO (ASiGs/PBS/DMSO). PBS containing 1% DMSO (PBS/DMSO) was used instead of ASiGs/PBS/DMSO for the control. ASiGs were dissolved in DMSO (200 μg/mL) and kept in the refrigerator as a stock solution before use. After one day, $5 \times 10^5$ cells of WT strain in 50 μL of PBS were intranasally injected into each mouse. After 3 days, mice were euthanized using carbon dioxide. Lung tissues from one mouse were homogenized in 3 mL of PBS using a bag homogenizer. Homogenates were centrifuged (13,500 × g for 10 min) to eliminate cell debris, and the concentration of each cytokine in the supernatant was quantified using ELISA (R&D SYSTEMS).

## Statistical analysis

Statistical analyses were conducted using GraphPad Prism version 6.05 or 6.07 for Windows (GraphPad Software). Data were analyzed using the unpaired two-tailed Student's t-test, one-way ANOVA with Tukey–Kramer test, two-way ANOVA with Fisher's least significant difference test, or log-rank test. Quantitative data are presented as Mean ± standard deviation (SD). Asterisks denote the level of significance (*, $P < 0.05$; **, $P < 0.01$; ***, $P < 0.001$). Significant differences between treatment groups were determined when $P$-values were < 0.05.

## Supporting information

**S1 Fig. Structures of EGCrP2 (A) and EGCase II (B) with each substrate.** A, Magnified view of the EG binding cleft in the EGCrP2-EG docking model. PDB code: 7LPO. EG can enter the substrate-binding cleft of EGCrP2/Sgl1; however, even lactosylceramide (LacCer), the minimum-sized substrate of EGCase, cannot enter the cleft of EGCrP2/Sgl1. B, Magnified view of the binding cleft of EGCase with GM3 ganglioside (*N*-acetylneuraminic acid-LacCer). PDB code: 5J7Z. LacCer and glycolipids with longer sugar chains than LacCer, such as GM3, can enter the cleft of EGCase by extending the sugar chain outward from the enzyme.
(TIF)

**S2 Fig. Phylogenetic tree of EGCrP2/Sgl1 and related proteins.** The amino acid sequences of EGCrP2/Sgl1 and related proteins were reconstructed using the neighbor-joining method. The scale bar indicates 0.5 amino acid

substitutions per site. The accession number of each protein is listed in S1 Table. Strains in the WHO FPPL critical group are shown in red.

(TIF)

**S3 Fig. Generation of *sgl1Δ::SGL1* (RE) strain.** A, *SGL1*-expressing construct was introduced into the *sgl1Δ* (KO) strain by targeted integration to be expressed under the control of its original promoter. B, *SGL1* KO construct was generated by the method described in [7]. C, PCR analysis of WT, KO, and RE strains. D, EGCrP2/Sgl1 activity of WT, KO, and RE strains. The activity was measured using C12-NBD-GlcCer as a substrate, as described previously [7]. Data are presented as Mean ± SD (n = 4).

(TIF)

**S4 Fig. Phloxine B staining of heat-killed *Cn* cells.** A, Phloxine B staining of control and heat-killed *Cn* cells. WT (top), KO (middle), and RE (bottom) *Cn* were incubated at 75°C (heat-killed) or 30°C (control) for 30 minutes before staining with phloxine B. Fluorescence image was captured using a fluorescence microscope with differential interference contrast (DIC) imaging. Scale bar, 25 µm. B, Spot assay of control and heat-killed *Cn* cells. Tenfold serial dilutions of each strain were spotted onto YPD agar plates and incubated at 30°C for 3 days.

(TIF)

**S5 Fig. Flow cytometry of WT, KO, and RE strains after staining with phloxine B.** WT, KO, and RE strains were cultured in YPD medium for 3, 4, and 7 days at 25, 30, and 37°C with shaking at 150 rpm. Cells were stained with phloxine B and analyzed by a flow cytometer using a BL2-H channel (574/26 nm BP filter). The data on day 3 was used as Fig 2D.

(TIF)

**S6 Fig. Clustering of genes in WT, KO and RE strains cultured at 30 and 37°C for 3 days.** A, K-means clustering of total genes and their expression patterns. RNAseq was performed using total RNA isolated from *Cn* after 72 h (day 3) of incubation at 30 and 37°C (n = 1 for each WT, KO, RE strains at both temperatures). The heat map shows the relative expression levels of each transcript (rows) in each sample (column). Normalized transcripts per million (TPM) were log 2-transformed and median-centered by transcript. The heatmap was drawn based on clustering results. Red and green colors represent higher and lower expressions, respectively. B, Enrichment scores of gene ontology (GO) terms for genes upregulated in KO strain in Cluster B on DAVID analysis [53]. The enrichment scores for each GO term are displayed in the graph.

(TIF)

**S7 Fig. Visualization of vacuoles by staining with FM4–64.** Vacuoles of GFP-Atg8 expressing WT, KO, and RE strains were stained with 16 µM of FM4–64 for 1 hour. A fluorescence microscope captured GFP (green) and F4-64 (red) images. Scale bar, 8 µm.

(TIF)

**S8 Fig. Subcellular localization of EGCrP2/Sgl1 of multiple cells of *Cn*.** Fluorescence microscopy images show mRuby-derived fluorescence (red), indicating the distribution of mRuby-tagged EGCrP2/Sgl1 (mRuby-EGCrP2/Sgl1) and free mRuby (mRuby). Vacuoles were stained with quinacrine (green), which specifically accumulates in fungal vacuoles [56]. Scale bar, 8 µm.

(TIF)

**S9 Fig. Accumulation of SGs and ASGs in EGCrP2/SglA-deficient *Aspergillus fumigatus*.** A, Generation of *SGLA* KO *A. fumigatus* (*sglaΔ*, KO) by split-marker method as described in [60]. B, Generation of *SGLA* revertant *A. fumigatus* (*sglaΔ::SGLA*, RE) by targeted integration. C, PCR analysis of WT, KO, and RE strains. D and E show the contents of SGs and ASGs, respectively, in conidia of WT, KO, and RE strains. *A. fumigatus* was cultured on YPD agar plates at

37°C, harvested, and disrupted using a bead crusher to obtain conidia. F and G show the contents of SGs and ASGs, respectively, in the hyphae of WT, KO, and RE strains. Hyphae of WT, KO, and RE strains were grown in YPD medium at 37°C, harvested, lyophilized, and subjected to analysis. Contents of SGs and ASGs extracted from conidia (corresponding to 1 μg of conidial protein) or 4 mg of dried hyphae were quantified using LC-ESI MS/MS. Data are presented as Mean ± SD (n = 3). DCW, dry cell weight.
(TIF)

**S10 Fig. Structures of chemically synthesized EG (A) and AEGs (B, C).** A, Ergosteryol β-D-glucopyranoside (EG); B, Ergosterol 6-*O*-palmitoyl-β-D-glucopyranoside (16:0-EG); C, Ergosterol 6-*O*-oleoyl-β-D-glucopyranoside (18:1-EG). The methods for synthesis of EG and AEGs are described in Materials and methods.
(TIF)

**S11 Fig. Docking models illustrating the interaction of mouse Mincle with AEG (18:1-EG) and EG.** A, B, AEG-docking model (18:1-EG, colored in green); C, D, EG-docking model (EG, colored in purple). The generation of docking models was performed according to the method described in Materials and methods. The Alphafold2 model of mouse Mincle was used in its apo state (lacking $Ca^{2+}$ and ligand binding) and was superimposed on the bovine Mincle structure (PDB ID: 5KTH) after truncating the first 62 N-terminal residues. In panels A and C, the surface of the Mincle is colored white, red, and blue to represent hydrophobic, negative, and positive regions, respectively.
(S11)

**S12 Fig. Identification of ABC transporter responsible for EG export in budding yeast.** A, Experimental design. B, Quantification of EG in the medium by LC-ESI MS/MS. The control strain (KO/OE) was generated by overexpression of *UGT51* [31] and disruption of *EGH1* [32] in *S. cerevisiae*. The test strain was generated from KO/OE by disrupting each ABC transporter. The *YOR1* revertant was generated by reintroducing *YOR1* into a *YOR1*-disrupted mutant (*yor1Δ*). Data are presented as Mean ± SD (n = 3).
(TIF)

**S13 Fig. MIP-2 production in BMDCs co-cultured with WT, KO, and RE strains.** BMDCs derived from WT and Mincle KO mice were co-cultured with *Cn* at a multiplicity of infection (MOI) of 1 and 5 in RPMI-1640 medium containing 10% FBS at 37°C under 5% $CO_2$ for 40 hours. The initial BMDC cell density was $1 \times 10^5$ cells/well. RPMI-1640 medium without penicillin and streptomycin was used to prevent the inhibitory effects of antibiotics. MIP-2 production was measured by ELISA, as described in Materials and methods. Data are presented as Mean ± SD (n = 3).
(TIF)

**S14 Fig. Cytokine production in BMDCs from wild type and Mincle KO mice by TDB and LPS.** MIP-2 (A) and TNF-α (B) productions in mouse BMDCs by TDB and LPS. BMDCs from WT and Mincle KO mice were exposed to TDB (0.1 nmol per well) and LPS (10 ng per well) at 37°C for 40 h. ELISA was performed to determine MIP-2 and TNF-α production using specific antibodies described in Materials and methods. Data are presented as Mean ± SD (n = 3).
(TIF)

**S15 Fig. Survival of WT and Mincle-deficient mice infected with WT and KO strains.** WT and Mincle KO mice (6 per group) were infected intranasally with $5 \times 10^5$ WT and KO strain cells. Mice were monitored twice daily, and those showing signs of distress or becoming moribund were humanely sacrificed using carbon dioxide. The experiments were independently repeated twice, confirming reproducibility. The combined results were subjected to statistical analysis.
(TIF)

**S16 Fig. Quality check of RNAs from WT, KO, and RE strains.** RNAs were prepared from WT, KO, and RE strains cultured in YPD medium at 30 and 37°C for 3 days following the methods described in Materials and methods. RNA integrity

was assessed using Agilent Bioanalyzer 2100 (Agilent Technologies), with RNA Integrity Numbers (RIN) measured for each sample. All samples had RIN values above 7, indicating that they were suitable for RNA sequencing and qPCR. (TIF)

**S1 Data. File containing the numerical data used to generate the figures (Fig 1A, Fig 1B, Fig 2A, Fig 2C, Fig 3, Fig 4B, Fig 5B, Fig 7B, Fig 8B, Fig 9E, Fig 9F, Fig 10A, Fig 10B, Fig 11B, Fig 11D, Fig 11E, Fig 11F, Fig 11G, Fig 11H, Fig 11I, Fig 13C, Fig 13D, Fig 13E, Fig 13F, Fig 14A, Fig 14B, Fig 14C, Fig 14D, Fig 14E, Fig 14F, Fig 14G, Fig 14H, Fig 15B, Fig 15C, Fig 15D, Fig 15E, Fig 15F, S3D Fig, S6A Fig, S6B Fig, S9D Fig, S9E Fig, S9F Fig, S9G Fig, S12B Fig, S13 Fig, S14A Fig, S14B Fig, S15 Fig, and S16 Fig).**
(XLSX)

**S1 Table. The accession number of EGCrP2/Sgl1 and related proteins.**
(XLSX)

**S2 Table. Oligonucleotide primers used in this study.**
(XLSX)

**S3 Table. MRM conditions for lipid analysis using LC-ESI MS/MS.**
(XLSX)

## Acknowledgments

We thank Ms. T. Honda, Ms. S. Uemura, Mr. S. Mochinaga, Dr. H. Goda, Ms. K. Urata, Ms. Y. Nakano, Dr. K. Toyonaga, Dr. S. Ishizuka, Mr. M. Kurata, Dr. S. Iwai, Mr. Y. Tsushima, Mr. R. Miyasaka, Ms. K. Ichikawa, Mr. R. Ugawa, and Mr. N. Iwachido for their technical assistance. We thank Dr. T. Kawamata at Institute of Science Tokyo (Japan) for her helpful advice and Dr. K. Kita at St. Francis College (USA) for reviewing the manuscript and providing valuable suggestions.

## Author contributions

**Conceptualization:** Takashi Watanabe, Yohei Ishibashi, Makoto Ito.

**Formal analysis:** Takashi Watanabe, Masayoshi Nagai, Yohei Ishibashi.

**Funding acquisition:** Sho Yamasaki, Makoto Ito.

**Investigation:** Takashi Watanabe, Masayoshi Nagai, Yohei Ishibashi, Mio Iwasaki, Masaki Mizoguchi, Masahiro Nagata, Takashi Imai, Koichi Takato, Akihiro Imamura, Yoshimitsu Kakuta, Takamasa Teramoto, Junko Matsuda.

**Project administration:** Makoto Ito.

**Resources:** Motohiro Tani, Hideharu Ishida, Sho Yamasaki.

**Supervision:** Makoto Ito.

**Validation:** Yohei Ishibashi, Nozomu Okino, Makoto Ito.

**Visualization:** Takashi Watanabe, Masayoshi Nagai, Yohei Ishibashi.

**Writing – original draft:** Takashi Watanabe, Masayoshi Nagai, Yohei Ishibashi, Yoshimitsu Kakuta, Makoto Ito.

**Writing – review & editing:** Takashi Watanabe, Masayoshi Nagai, Yohei Ishibashi, Makoto Ito.

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
