## [Decision Letter · Decision Letter 0]

24 Nov 2024

PPATHOGENS-D-24-02306Vacuolar sterol β-glucosidase EGCrP2/Sgl1 deficiency in *Cryptococcus neoformans* : dysfunctional autophagy and Mincle-dependent immune activation as targets of novel antifungal strategiesPLOS Pathogen Dear Dr. Ito, Thank you for submitting your manuscript to PLOS Pathogens. After careful consideration, we feel that it has merit but does not fully meet PLOS Pathogens's publication criteria as it currently stands. Therefore, we invite you to submit a revised version of the manuscript that addresses the points raised during the review process Please submit your revised manuscript within 60 days Jan 23 2025 11:59PM. If you will need more time than this to complete your revisions, please reply to this message or contact the journal office at plospathogens@plos.org. Please include the following items when submitting your revised manuscript: * A rebuttal letter that responds to each point raised by the editor and reviewer(s). You should upload this letter as a separate file labeled 'Response to Reviewers '. This file does not need to include responses to any formatting updates and technical items listed in the 'Journal Requirements' section below. * A marked-up copy of your manuscript that highlights changes made to the original version. You should upload this as a separate file labeled 'Revised Manuscript with Track Changes '. * An unmarked version of your revised paper without tracked changes. You should upload this as a separate file labeled 'Manuscript '. If you would like to make changes to your financial disclosure, competing interests statement, or data availability statement, please make these updates within the submission form at the time of resubmission. Guidelines for resubmitting your figure files are available below the reviewer comments at the end of this letter. We look forward to receiving your revised manuscript. Kind regards, Kirsten Nielsen, Ph.DGuest EditorPLOS Pathogens Michal OlszewskiSection EditorPLOS Pathogens Michael Malim

Editor-in-Chief

PLOS Pathogens

orcid.org/0000-0002-7699-2064 **Additional Editor Comments :** The concerns regarding controls raised by Reviewer 2 and 3 need to be addressed. Both reviewers 1 and 2 were concerned about the biological relevance of the studies and these concerns should be addressed with proposed experiments or similar ones that provide the necessary clarification. **Journal Requirements:**

At this stage, the following Authors/Authors require contributions: Takashi Watanabe, Masayoshi Nagai, Yohei Ishibashi, Mio Iwasaki, Masaki Mizoguchi, Masahiro Nagata, Takashi Imai, Koichi Takato, Akihiro Imamura, Yoshimitsu Kakuta, Takamasa Teramoto, Motohiro Tani, Junko Matsuda, Hideharu Ishida, Sho Yamasaki, Nozomu Okino, and Makoto Ito. Please ensure that the full contributions of each author are acknowledged in the "Add/Edit/Remove Authors" section of our submission form.

3) Your study includes live participants, but you have not included an Ethics Statement. Please include an Ethics Statement subsection to your Materials and Methods, to include:

The name(s) of the Institutional Review Board(s) or Ethics Committee(s)ii) The approval number(s), or a statement that approval was granted by the named board(s)iii) (for human participants or donors) - A statement that formal consent was obtained (must state whether verbal/written) OR the reason consent was not obtained (e.g., anonymity).

Potential Copyright Issues:

i) Please confirm (a) that you are the photographer of figure 1C, or (b) provide written permission from the photographer to publish the photo(s) under our CC BY 4.0 license.

6) In the online submission form, you indicated that " all data supporting this study's results are available in the article and its Supplementary Appendix or from the corresponding authors upon reasonable request." All PLOS journals now require all data underlying the findings described in their manuscript to be freely available to other researchers, either

1. In a public repository

2. Within the manuscript itself

3. Uploaded as supplementary information.

7) Please amend your detailed Financial Disclosure statement. This is published with the article. It must therefore be completed in full sentences and contain the exact wording you wish to be published.

2) State what role the funders took in the study. If the funders had no role in your study, please state: "The funders had no role in study design, data collection and analysis, decision to publish, or preparation of the manuscript.".

8) We note that your Data Availability Statement is currently as follows:" all data supporting this study's results are available in the article and its Supplementary Appendix." Please confirm at this time whether or not your submission contains all raw data required to replicate the results of your study. Authors must share the “minimal data set” for their submission. PLOS defines the minimal data set to consist of the data required to replicate all study findings reported in the article, as well as related metadata and methods (https://journals.plos.org/plosone/s/data-availability#loc-minimal-data-set-definition). For example, authors should submit the following data: - The values behind the means, standard deviations and other measures reported; - The values used to build graphs; - The points extracted from images for analysis. Authors do not need to submit their entire data set if only a portion of the data was used in the reported study. If your submission does not contain these data, please either upload them as Supporting Information files or deposit them to a stable, public repository and provide us with the relevant URLs, DOIs, or accession numbers. For a list of recommended repositories, please see https://journals.plos.org/plosone/s/recommended-repositories. If there are ethical or legal restrictions on sharing a de-identified data set, please explain them in detail (e.g., data contain potentially sensitive information, data are owned by a third-party organization, etc.) and who has imposed them (e.g., an ethics committee). Please also provide contact information for a data access committee, ethics committee, or other institutional body to which data requests may be sent. If data are owned by a third party, please indicate how others may request data access.

**Reviewers' Comments:**Reviewer's Responses to Questions

**Part I - Summary**

Reviewer #1: In this study from Watanabe et al, the authors aim to understand the mechanisms underlying decreased virulence that has been previously observed in Cryptococcus neoformans strains deficient for sterol-beta-glucosidase. The authors suggest a two-fold mechanism: 1) Sgl1-deficient Cn results in increased rates of yeast cell death when cultured at 37C due to defective autophagy, and 2) Deletion of Sgl1 results in accumulation of ergosterol-beta-glucoside as wells as acetylated ergosterol-beta-glucoside, the latter of which can function as a ligand for the C-type lectin receptor Mincle when secreted by EVs. Overall, the results demonstrating defective autophagy in vitro and accumulation of EGs and AEGs are clear. The identification of AEGs as Mincle ligands is also an important insight.

However, the significance of the autophagy defect vs Mincle activation in vivo, and the actual activation of Mincle by the KO Cn strain are less convincing. The authors generally rely on ultracentrifugation of supernatants to isolate EVs and then test their ability to activate DCs, however, this approach may result in non-physiological enrichment of trace amounts of lipid activity.

Major Comments:

1. The authors show clearly that 37C culture of KO Cn results in increased yeast cell death and that this correlates with decreased expression of autophagy genes and localization of ATG8 with vacuoles. What is less clear to me is whether the observed effects on autophagy are a cause or a consequence of the cell death at 37C? Much of the analyses done for testing autophagy function are done in nitrogen starvation media – how does this relate to the late increase in cell death when culturing in YPD? Are there ways to rescue/increase autophagy in the KO cells to see if this rescues?

2. I’m convinced that KO Cn produces more EG and AEGs, and that AEGs will activate Mincle when you add them to BMDCs or into reporter lines. My concern with the EV experiments is that these may be added in supraphysiological concentrations in the functional assays. In Fig 12, the KO EVs give only a 2-fold increase in production of MIP-2 from BMDCs compared with WT, which is inconsistent with almost non-existent concentrations of EG and AEGs in WT Cn compared to KO. As a very simple experiment, if you just challenge BMDCs or the Mincle reporter line with WT vs KO Cn, do the KO cells actually secrete enough AEG to activate? If so, can you observe the same effect with conditioned supernatants? These sorts of experiments are crucial to perform before doing ultracentrifugation on supernatants.

3. Can the authors perform some experiments to assess the relative contributions of autophagy defects vs increased Mincle production to the virulence defect in vivo? The authors show an increase in CFU of KO Cn at Day 3 post-infection in Mincle-deficient mice compared to WT mice. This is a crucial line of experiments in that it asks how much of the in vivo defect is due to increased Mincle ligand production vs autophagy defects – I think the manuscript would be well-served to do this experiment as a survival curve or CFU time course to see how much the infection outcome is altered in Mincle KO mice. Alternatively, since Card9 deletion has a much stronger phenotype in vitro for loss of activation in response to AEGs, it might be better to see how much of the CFU defect of Sgl1KO Cn is rescued in Card9-/- mice.

4. Along the same lines as above, the ASG treatment along with WT Cn infection is a nice experiment in that it tests how much of an effect Mincle activation alone has on CFU in the absence of an autophagy defect. However, the actual result is not very convincing. Was this just done once? This experiment warrants a repeat and looking at later time points (if you give repeated ASG doses).

Minor comments:

1. The OD600 defects actually seem more pronounced at 25C and 30C (at least at day 7) compared to 37C. The authors suggest that this may be due to leakage of cellular contents at 37C. However, they also observe very small increased in dead Cn at 25 and 30, despite the strong OD600 defect. Can they comment on this?

2. For the model in Fig 14, I understand some of this is linking in previously published data. However, for this specific manuscript, I think the authors should remove the images from the model that are not specifically based on experiments shown in this body of work. For example, the link to gamma delta T cells is entirely speculative.

Reviewer #2: In general, this paper is a succession of explorations related to the EGCrP2/Sgl1 KO mutant. I fell that there is not much link between the different parts. For example, the autophagy is not really related to the other explorations such as immunology and then EVs. We are missing here a united line for the message of this paper.

Reviewer #3: In this very nice study, the authors investigated the mechanisms responsible for the loss of virulence of a slg1delta mutant in Cryptococcus neoformans. This gene has been identified by this group and that of M. del Poeta, and this deletion has been found to be associated with the accumulation of ergosterol beta-glucoside (EG), but the reason for the loss of virulence remained unknown. In this paper, they showed that Slg1 accumulates in the vacuole and controls autophagy gene expression and protein localization. The authors used a battery of assays to show that a concomitant accumulation of EG acetylated derivatives (AEGs) is probably more important than that of EG. They presented convincing experiments showing that AEGs bind to the C-type lectin receptor Mincle in a very affine manner, activating the production of cytokines that ultimately trigger the elimination of the yeast pathogen. They confirmed the importance of both this lipid and the receptor using animal models coupled with AEGs treatments. It is a very nice story and a very convincing set of data.

**Part II – Major Issues: Key Experiments Required for Acceptance**

Reviewer #1: 1. Challenge WT vs Mincle-/- vs Card9-/- BMDCs with WT vs KO Cn live yeasts to see if the KO strain produces enough AEGs to activate the Mincle pathway.

2. Infect Mincle-/- and/or Card9-/- mice with WT vs KO Cn and perform either survival curves or time course (day 3, 7, 10, 14) CFUs

3. Infect WT vs Mincle-/- mice with WT Cn along with ASGs and perform time course CFUs.

Reviewer #2: All the experiments are performed after 3 days in YPD with agitation. The cells are at this step starved and so not in a normal metabolism. Is it really physiological and do the findings could be generalized to LOG or stationary phase?

How far Phloxin B have been validated in Cryptococcus neoformans we need controls together with the experiment (Dead control 65°C 1h, H2O2 2 to 24hrs).

RNASeq have been perform as simplicates per condition with prevent a accurate analysis of the results in particular normalization and adequate calculation of fold changes. DeSeq 2 in the best method/pipeline to analyze RNASeq and qhould be done like this.

Relative expression by qPCR should not use 2-∆∆Cq as individual qPCR have a different efficiency and should use at least 3 reference genes (see 1. Vandesompele J, Preter K de, Pattyn F, Poppe B, Roy N van, Paepe A de, Speleman F. Accurate normalization of real-time quantitative RT-PCR data by geometric averaging of multiple internal control genes. Genome biology. 2002 Jun 18;3(7):RESEARCH0034. )

Considering that more than 60% of the cell population is dead (Fig 2C), it is difficult to make sens of the transcriptome analysis of the cells cultivated in the same condition. Are the ATG8 transcript decreased because of death? I would assume so.Probably the half-life of those transcript are much lower than that of the reference gene explaining this decrease Iin the expression.

Enrichment score is between 1 and 1.5 which is not high. Then, is it really meaningful?

I am not sure that the finding on fumigatus are really bringing something to the message on cryptococcus (Aspergillus ascomycetes and Cryptococcus basitiomycetes are indeed very different organisms in terms of biology.)

Reviewer #3: The following comments are intended to improve the quality of this manuscript.

1) The use of Phloxin B staining to determine cell viability needs to be validated. In the image shown (Figure 2B), both mother and budding cells are stained, which is strange for a dying cell.

2) Figure 5 shows not only degradation but also production of Atg8, which according to the qPCR experiments also decreases over time. This needs to be better explained.

3) Sgl1 localization (Figure 6): showing one cell at high magnification is good, but it should be accompanied by an image showing multiple cells to convince the reviewer that this co-localization is representative.

4) Line 280 "significantly higher should be confirmed by a statistical test and a p-value should be given.

5) The paragraph starting on 309 describing the mouse mincle docking models with AEG or AG is interesting, but the conclusion would need more experiments to be confirmed. The calcium implication comes out of nowhere for the reader. Its implication and role could be confirmed experimentally. This paragraph could be deleted as well.

6) Same for the next paragraph. The point mutation mutant should be tested if one wants to reach a conclusion.

7) Figure 11C, E, F: EG and AEG are detectable in WT and RE strain. This is surprising as it was stated earlier that it is not detectable in the WT cell. This should be discussed.

8) Line 368: There is probably a word missing in this sentence and in the next one, since EVs do not produce MIP-2 or IL-8.

**Part III – Minor Issues: Editorial and Data Presentation Modifications**

Reviewer #1: N/A

Reviewer #2: (No Response)

Reviewer #3: none

PLOS authors have the option to publish the peer review history of their article (what does this mean? ). If published, this will include your full peer review and any attached files.

**Do you want your identity to be public for this peer review?** For information about this choice, including consent withdrawal, please see our Privacy Policy .

Reviewer #1: No

Reviewer #2: No

Reviewer #3: No

**Figure resubmission:**While revising your submission, please upload your figure files to the Preflight Analysis and Conversion Engine (PACE) digital diagnostic tool, https://pacev2.apexcovantage.com/. PACE helps ensure that figures meet PLOS requirements. To use PACE, you must first register as a user. Registration is free. Then, login and navigate to the UPLOAD tab, where you will find detailed instructions on how to use the tool. If you encounter any issues or have any questions when using PACE, please email PLOS at figures@plos.org. Please note that Supporting Information files do not need this step. If there are other versions of figure files still present in your submission file inventory at resubmission, please replace them with the PACE-processed versions. **Reproducibility:**To enhance the reproducibility of your results, we recommend that authors of applicable studies deposit laboratory protocols in protocols.io, where a protocol can be assigned its own identifier (DOI) such that it can be cited independently in the future. Additionally, PLOS ONE offers an option to publish peer-reviewed clinical study protocols. Read more information on sharing protocols at https://plos.org/protocols?utm_medium=editorial-email&utm_source=authorletters&utm_campaign=protocols

---

## [Decision Letter · Decision Letter 1]

5 Mar 2025

PPATHOGENS-D-24-02306R1

Vacuolar sterol β-glucosidase EGCrP2/Sgl1 deficiency in *Cryptococcus neoformans* : dysfunctional autophagy and Mincle-dependent immune activation as targets of novel antifungal strategies

PLOS Pathogens

Dear Dr. Ito,

Thank you for submitting your manuscript to PLOS Pathogens. After careful consideration, we feel that it has merit but does not fully meet PLOS Pathogens's publication criteria as it currently stands. Therefore, we invite you to submit a revised version of the manuscript that addresses the points raised during the review process. Specifically: 1) Move the RNAseq figure(s) to supplemental data and clearly state in the manuscript that the studies were preliminary and had only a single replicate. 2) Address Reviewer 3's concerns regarding the validity of the results from the PhloxinB results based on the current studies. An additional experiment may be warranted.

Please submit your revised manuscript within 60 days May 04 2025 11:59PM. If you will need more time than this to complete your revisions, please reply to this message or contact the journal office at plospathogens@plos.org. Please include the following items when submitting your revised manuscript:

We look forward to receiving your revised manuscript.

Kind regards,

Kirsten Nielsen, Ph.D

Guest Editor

PLOS Pathogens

Michal Olszewski

Section Editor

PLOS Pathogens

 Sumita Bhaduri-McIntosh

Editor-in-Chief

PLOS Pathogens

orcid.org/0000-0003-2946-9497

 Michael Malim

Editor-in-Chief

PLOS Pathogens

orcid.org/0000-0002-7699-2064

**Journal Requirements:**

3) Please ensure that the funders and grant numbers match between the Financial Disclosure field and the Funding Information tab in your submission form. Note that the funders must be provided in the same order in both places as well.

**Reviewers' Comments:**

Reviewer's Responses to Questions

**Part I - Summary**

Reviewer #1: The authors have satisfactorily addressed my previous comments.

Reviewer #3: The paper has been improved, but there are still a few issues that need to be addressed.

**Part II – Major Issues: Key Experiments Required for Acceptance**

Reviewer #1: N/A

Reviewer #3: 1) The experiment to validate the PhloxinB assay as a means of assessing cell viability is not the one expected. The author should sort out the PhloxinB positive cells and plate them to check that they are dead. Again, the fact that both the mother cell and the bud can be positive suggests that the assay is not valid.

2) I missed this in my first review, but having a single replicate for the RNA-Seq experiment is a real problem. I understand that the authors see this experiment as a way to identify potentially regulatory genes that would be validated by qRT-PCR. However, the description of the RNA-Seq result should be different. For example, it is possible to write and RNA-seq data analysis 'revealed' something. This paragraph should be changed and the fact that a single replicate was generated should be clearly stated in the results paragraph. The figure 3A and B should be deleted.

**Part III – Minor Issues: Editorial and Data Presentation Modifications**

Reviewer #1: N/A

Reviewer #3: 1) The name of the gene should be changed. The official name is SLG1, as should be a gene name in Cryptococcus (3 letters and a number). It cannot be EGCrP2/Sgl1. This should be changed in the manuscript.

2) The strains cannot be named KO Cn or RE Cn. slg1delta should identify the mutant strain. The reconstituted strain should be named slg1delta::SLG1.

PLOS authors have the option to publish the peer review history of their article (what does this mean? ). If published, this will include your full peer review and any attached files.

**Do you want your identity to be public for this peer review?** For information about this choice, including consent withdrawal, please see our Privacy Policy .

Reviewer #1: No

Reviewer #3: No

**Figure resubmission:**
---

## [Editor Report · Decision Letter 2]

28 Mar 2025

Dear Dr. Ito,

We are pleased to inform you that your manuscript 'Vacuolar sterol β-glucosidase EGCrP2/Sgl1 deficiency in *Cryptococcus neoformans* : dysfunctional autophagy and Mincle-dependent immune activation as targets of novel antifungal strategies' has been provisionally accepted for publication in PLOS Pathogens.

Best regards,

Kirsten Nielsen, Ph.D

Guest Editor

PLOS Pathogens

Michal Olszewski

Section Editor

PLOS Pathogens

Sumita Bhaduri-McIntosh

Editor-in-Chief

PLOS Pathogens

orcid.org/0000-0003-2946-9497

Michael Malim

Editor-in-Chief

PLOS Pathogens

orcid.org/0000-0002-7699-2064

The authors adequately responded to the reviewer and editor comments.
---

## [Editor Report · Acceptance letter]

Dear Dr. Ito,

We are delighted to inform you that your manuscript, "Vacuolar sterol β-glucosidase EGCrP2/Sgl1 deficiency in *Cryptococcus neoformans* : dysfunctional autophagy and Mincle-dependent immune activation as targets of novel antifungal strategies," has been formally accepted for publication in PLOS Pathogens.

Best regards,

Sumita Bhaduri-McIntosh

Editor-in-Chief

PLOS Pathogens

orcid.org/0000-0003-2946-9497

Michael Malim

Editor-in-Chief

PLOS Pathogens

orcid.org/0000-0002-7699-2064